Direct and semi-direct radiative forcing of biomass burning aerosols over the Southeast Atlantic (SEA) and its sensitivity to absorbing properties: a regional climate modeling study.

Marc Mallet[1], Fabien Solmon[2], Pierre Nabat [1], Nellie Elguindi[2], Fabien Waquet[3], Dominique Bouniol[1], Andrew Mark Sayer[4,5], Kerry Meyer[5], Romain Roehrig[1], Martine Michou[1], Paquita Zuidema[6], Cyrille Flamant[7], Jens Redemann[8] and Paola Formenti[9]

[1] Centre National de Recherches Météorologiques, UMR3589, Météo-France-CNRS, Toulouse, France

[2] Laboratoire d'Aérologie, UMR 5560, 16 avenue Édouard Belin, 31400 Toulouse, France

[3] Université de Lille, CNRS, UMR 8518, LOA - Laboratoire d'Optique Atmosphérique, 59000 Lille, France

[4] Universities Spcae Reasearch Association Columbia, MD, USA

[5] NASA Goddard Spade Flight Center, Greenbelt, MD, USA

[6] Rosenstiel School of Marine and Atmospheric Sciences, University of Miami, Miami, FL, USA

[7] LATMOS/IPSL, Sorbonne Université, UVSQ, CNRS, Paris, France

[8] University of Oklahoma, Norman, Oklahoma, USA

[9] LISA, UMR CNRS 7583, Université Paris Est Créteil et Université Paris Diderot, Institut Pierre Simon Laplace, Créteil, France

**Abstract**

Simulations are performed for the period 2000-2015 by two different regional climate models, ALADIN and RegCM, to quantify the direct and semi-direct radiative effects of biomass burning aerosols (BBA) in the Southeast Atlantic (SEA) region. Different simulations have been performed using strongly absorbing BBA in accordance with recent in situ observations over the SEA. For the July-August-September (JAS) season, the single scattering albedo (SSA) and total aerosol optical depth (AOD) simulated by the ALADIN and RegCM models are consistent with the MACv2 climatology and MERRA-2 and CAMS-RA reanalyses near the biomass burning emission sources. However, the above-cloud AOD is slightly underestimated compared to satellite (MODIS and POLDER) data during the transport over the SEA. The direct radiative effect exerted at the continental and oceanic surfaces by BBA is significant in both models and the radiative effects at the top of the atmosphere indicate a remarkable regional contrast over SEA (in all-sky conditions), with a cooling (warming) north (south) of 10°S, which is in agreement with the recent MACv2 climatology. In addition, the two models indicate that BBA are responsible for an important shortwave radiative heating of ~0.5-1 K per day over SEA during JAS with maxima between 2 and 4 km above mean sea-level. At these altitudes, BBA increase air temperature by ~0.2-0.5 K, with the highest values being co-located with low stratocumulus clouds. Vertical changes in air temperature limit the subsidence of air mass over SEA creating a cyclonic anomaly. The opposite effect is simulated over the continent due to the increase in lower troposphere stability. The BBA semi-direct effect on the lower troposphere circulation is found to be consistent between the two models. Changes in the cloud fraction are moderate in response to the presence of smoke and the models differ over the Gulf of Guinea. Finally, the results indicate an important sensitivity of the direct and semi-direct effects to the absorbing properties of BBA. Over the Sc region, DRE varies from +0.94 W m$^{-2}$ (scattering BBA) to +3.93 W m$^{-2}$ (most absorbing BBA).

## 1. Introduction

In addition to their direct radiative effect (DRE), solar radiation absorbing aerosols (AA), such as biomass burning aerosol (BBA) from vegetation fires and mineral dust from aeolian erosion of arid and semi-arid soils, are known to affect regional and global climate through the semi-direct effect (SDE) (Ackerman et al. 2000). The SDE is initiated by modifications in the vertical profile of the shortwave radiative heating and atmospheric temperature due to the absorption of solar radiation by AA. Such perturbations in the lower troposphere radiative budget can impact atmospheric vertical stability, circulation and cloud properties. This radiative effect is extremely sensitive to the AA load and vertical distribution in the atmosphere, especially in the presence of cloud layers (Koch and Del Genio., 2010). For instance, AA can increase the water content of low-level clouds, particularly when AA are transported above the cloud layer, by stabilizing the free troposphere and increasing the strength of the temperature inversion capping the cloud top, decreasing dry-air entrainment into the low-level clouds (Johnson et al., 2004; Wilcox, 2010, Deaconu et al., 2019, Herbert et al., 2020). Contrarily, when AA are in contact with low-clouds, they may decrease low-cloud cover by heating the air and reducing relative humidity (Hansen et al., 1997, Ackerman et al., 2000).

At the global scale, Perlwitz and Miller (2010) have indicated an increase of low cloud cover due to mineral dust with increasing aerosol absorption. In addition, results from the Precipitation Driver Response Model Intercomparison Project (PDRMIP) have shown that a tenfold increase in black carbon (BC) leads to a robust increase in globally averaged low-level clouds and to a reduction in mid-level and high-level clouds (Stjern et al., 2017). Contrarily, based on different global climate models, Allen et al. (2019) find an opposite effect, where a global annual mean decrease in low and mid-level clouds is associated with weaker decreases in high-level clouds, implying that cloud adjustments act to warm the climate system. Regionally, this study also highlights an important multi-model response found over Southern Africa, in which high and low-level clouds are significantly increased over the continent. In this region, Sakaeda et al. (2011) provided model estimates of regional radiative forcing from direct and semi-direct effects, which has significant impacts on cloud properties by increasing low cloud cover, notably over the ocean. Randles and Ramaswamy (2010) have also examined the direct and semi-direct impacts of absorbing biomass burning aerosol on the climate of southern Africa using an atmospheric general circulation model. The authors indicate that strong atmospheric absorption from these particles can cool the surface and increase upward motion and low-level convergence over southern Africa during the dry season.

AA can also impact regional or global atmospheric circulation. In Western Africa, Lau et al. (2009) argue that absorbing dust can trigger the Elevated Heat Pump effect, impacting the African monsoon dynamics and Sahel precipitation. In the same region, Solmon et al. (2008, 2012) also demonstrated the sensitivity of monsoon dynamics and precipitation to AA (mineral dust) optical properties. Several studies conducted during the Indian Ocean Experiment (INDOEX) and Aerosol Characterization Experiment (ACE)-Asia projects have also demonstrated that polluted aerosols containing BC could affect the regional circulation and hydrological cycle over the Indian and Asian regions (Ramanathan et al., 2000; Lau et al. 2006; Bollasina et al., 2014). These changes have also been found to be strongly related to the absorbing vs diffusive nature of anthropogenic aerosols. Over tropical Africa, Tosca et al. (2015) indicate a reduction in cloud fraction during periods of high aerosol optical depths related to a smoke-driven inhibition of convection.

BBA represent one of the main aerosol species able to induce a significant SDE at regional and global scales. Due to the large fraction of BC within the smoke plumes, BBA absorb SW radiation and are characterized by a single scattering albedo (SSA) significantly lower than unity (Dubovik et al., 2002). From Aerosol Robotic Network (AERONET) retrievals in Zambia, Eck et al. (2013) reported SSA between 0.80 and 0.86 (at 550 nm) during the biomass burning

season, with minima in July. During SAFARI-2000 (South Africa), Leahy et al. (2007) indicate a « campaign-average » SSA (550 nm) of 0.85 ± 0.02. Over Western Africa, Johnson et al. (2008) reported SSA from 0.73 to 0.93 (550 nm) in aerosol layers dominated by biomass burning during the Dust and Biomass-burning Experiment (DABEX) campaign, while values of 0.79 and 0.88 have been obtained over different regions in South America (Darbyshire et al., 2019). Over the SEA, Pistone et al. (2019) report that the ORACLES-2016 measured or retrieved SSA (at 500 nm) ranges between 0.85 and 0.88, depending on the instrument used.

Interestingly, recent observations obtained during the LASIC project (Zuidema et al., 2016) measured extremely low SSA (~0.75 at 550 nm) for aged BBA at Ascension Island (Zuidema et al., 2018), similar to values reported by Denjean et al. (2020) for smoke aerosols transported over the Gulf of Guinea during the DACCIWA experiment (Flamant et al., 2018). Such low values are consistent with recent findings obtained during the Clouds and Aerosol Radiative Impacts and Forcing CLARIFY project (Wu et al., 2020). The possible mixing state (external/internal) of BC particles contained within smoke plumes, combined with photochemical oxidation (Wu et al., 2020) and loss of organic aerosol during transport, represent possible processes explaining such low values. These recent outstanding absorbing properties of BBA measured over the SEA, associated with the important loading of smoke particles transported above Sc in the SEA (Sayer et al., 2019, Kacenelenbogen et al., 2019, Mallet et al., 2019) could have important implications in terms of direct and semi-direct radiative effect. Quantifying these impacts and related feedbacks at the climatic time scale is one of the main objectives of the present study.

Until now, most studies have focused on specific events. For example, Lu et al. (2018) quantified an average SDE plus DRE of -1.0 W.m$^{-2}$ for a two-month large eddy simulation over SEA, which is significantly smaller than the indirect forcing (-7.0 W.m$^{-2}$). Gordon et al. (2018) investigated a 10-day case study during August 2016 using the HadGEM global climate model at convection-permitting spatial resolution. They indicate a substantial positive DRE (+11 W m$^{-2}$) at the regional scale associated with important SDE (−30 W m$^{-2}$) and indirect forcing (−10 W m$^{-2}$). In that study, the microphysical and dynamical changes led to an increase in liquid water path (LWP) relative to a simulation without BBA. Finally, recent field measurements obtained at Ascension Island reveal that the low cloud fraction (LCF) decreases with enhanced smoke loadings within the boundary layer, suggesting a positive feedback of SDE (Zhang and Zuidema, 2019). To our knowledge, Sakeada et al. (2011) and Allen et al. (2019) are the only studies which have investigated the DRE/SDE of BBA at a climatic scale using global atmospheric models.

This study investigates these radiative effects over SEA at a climatic scale. Two independent regional climate models (RCMs) are employed for assessing the robustness of the results. We specifically investigate the SDE of BBA on the dynamics of the lower troposphere over SEA for the period 2000-2015, as well as the induced changes on low-cloud properties. We also propose the first set of long-term simulations of both DRE and SDE using extreme absorbing properties of BBA based on recent in situ observations (Zuidema et al., 2018; Denjean et al., 2020; Wu et al., 2020) obtained over the tropical African region. In this context, the main scientific questions are the following:

- What is the shortwave DRE of BBA at the surface and at the top of the atmosphere (TOA) in all-sky conditions over SEA and Central Africa?

- How much is the induced SW heating of BBA and what are its impact on the atmospheric temperature profile?

- What is the impact of the SDE of BBA on the lower troposphere circulation and Sc properties?

- What is the sensitivity of DRE and SDE to smoke absorbing properties?

To address these scientific questions, this study is organized as follows: Section 2 describes the different simulations and the data sets used for the model evaluation. Section 3 evaluates the representation of the SEA mean climate, as well

as Sc and BBA optical properties. Section 4 and 5 quantify respectively the DRE (at the surface and TOA) and SDE (on the lower troposphere atmospheric circulation and low-cloud properties) of smoke particles, respectively. Finally, Section 6 investigates the sensitivity of both forcing to BBA absorbing properties. Conclusions are given in Section 7.

## 2. Methodology

2.1 Models and Simulations

2.1.1 ALADIN and RegCM

This study relies on two regional climate models, namely CNRM-ALADIN63 and RegCM, described by Nabat et al. (2020) and Giorgi et al. (2012), respectively. Both models are driven by the ERA-Interim (ERAI) reanalysis over a period covering 2000-2015 (ALADIN) and 2003-2015 (RegCM). Sea Surface Temperatures (SSTs) are prescribed for ALADIN, whereas RegCM uses a slab ocean approach described in Solmon et al. (2015). It should be noted that prescribed SST can also be altered by the aerosol radiative effect. Different domains and spatial resolutions have been considered (see Table 1). ALADIN uses a 12 km horizontal resolution with 91 vertical levels (from 1015 to 0.01 hPa), focusing on a Southern Africa domain, while RegCM uses an 80 km horizontal resolution (with 42 vertical levels up to 50 hPa, see Table 1) on a large pan-African domain (latitude: -35°S to 30°N; longitude: -30°W to 45°E). In ALADIN, the possible long-range transport of BBA is not forced at the lateral boundary conditions, but the rather large domain (latitude: -37.1°S to 09.4°N; longitude: -33.4°W to 45.4°E) encompasses the main biomass burning sources. Land surface processes are treated using the SURFEX (Surface Externalisée) model (Masson et al., 2013; Decharme et al., 2019). In RegCM, chemical boundary conditions are given by monthly aerosol fields derived from an EC-EARTH-CAMS global simulation. CLM45 is used as the land surface scheme and the Tiedke scheme for convection. Of primary importance, we use the University of Washington planetary boundary layer turbulence scheme, which has been evaluated over the Californian region by O'Brien et al. (2012), showing a notable improvement in the representation of low Sc. The rapid radiative transfer model (RRTM) radiative transfer scheme is used to calculate interactions between aerosol radiative properties and shortwave and longwave radiation (for coarse dust and sea-salt particles).

Finally, the statistical cloud parametrization used in ALADIN is based on the work of Sommeria and Deardorff (1977) and Bougeault (1981) and coupled to the turbulence scheme (Cuxart et al., 2000) to derive subgrid-scale variances. This is fully described in Roehrig et al. (2020). In RegCM, the convective cloud fraction is parametrized according to selected convective schemes, while cloud water content is estimated depending on a temperature based parametrisation (Giorgi et al., 2012). Subgrid cloud fractions and cloud water content are combined to resolved cloud fraction and water content before being passed to the radiation scheme.

### 2.1.2 Aerosol schemes

The aerosol schemes of the two models are quite similar in terms of complexity and compatible with climate scale integrations. In ALADIN, the TACTIC (Tropospheric Aerosols for ClimaTe in CNRM) aerosol scheme accounts for sulfate, organic (OC) and black (BC) carbon, dust and primary sea-salt particles (Nabat et al., 2015; Michou et al., 2015, 2019, Mallet et al., 2019). In RegCM, the option used here is described in Solmon et al. (2006), Tummon et al. (2010) and Malavelle et al. (2011), with a special treatment for biomass burning aerosol described through a "smoke" tracer as described in Section 2.1.3. In both models, mineral dust and sea-salt emissions are interactively connected with surface meteorological fields and soil properties (Nabat et al., 2015; Solmon et al., 2008, 2012). The emission of mineral dust is primarily taken into account following Marticorena and Bergametti (1995), while the current formulation for primary sea spray is based on Grythe et al. (2014) for ALADIN and Zakey et al. (2008) for RegCM. These models include tracer advection by atmospheric winds, diffusion by turbulence and surface emissions, as well as

dry and wet (in-cloud and below-cloud) removal processes. In both RCMs, a bulk approach is applied for primary BC, OC and sulfate, whereby a fixed aerosol size distribution is assumed for calculating aerosol properties. In the two models, a more resolved size distribution (6 or 12 fixed bins) is used for primary mineral dust and sea-salt particles.

Both models assume external mixing of the different aerosol species, which could potentially be a limitation, especially with regard to possible OC/BC mixing (internal/external) state, which can significantly affect SW absorption (Fierce et al., 2016). Knowing that, specific attention is given to the evaluation of the simulated single scattering albedo of BBA in this study. The radiative properties (mass extinction efficiency, SSA, and asymmetry parameter) of each aerosol species
are calculated for the different spectral bands of the Fouquart and Morcrette radiation scheme (FMR; Morcrette, 1989) and the Rapid Radiative Transfer Model (RRTM; Mlawer et al., 1997) for SW and longwave (LW) radiation respectively, in ALADIN-Climat, and RRTM for RegCM (see Table 1). Aerosol forcing at the surface and TOA in SW and LW spectral ranges, in both clear-sky/all-sky conditions, are diagnosed using a double call to the radiation schemes
during the model integration. The DRE is calculated following Ghan et al. (2013).

### 2.1.3 Representation of BBA

Following Mallet et al. (2017, 2019), two tracers have been implemented in both regional models describing the mass concentration of fresh (less hygroscopic) and aged (more hygroscopic) smoke aerosols. This method allows

distinguishing between aerosols from biomass burning and anthropogenic emissions and to monitor specific properties, such as e-folding time, hygroscopic and optical properties. Although many GCMs represent BBA as separate components (BC and OC), this approach allows the representation of BBA as a single species including fresh and aged modes, making comparisons using aircraft and remote-sensing observations that characterize the ambient BBA rather
than BC and OC components more straightforward. With this approach, the BBA aerosol model properties can still be adjusted and/or evaluated using regional experimental campaigns over SEA such as ObseRvations of Aerosols above Clouds and their intEractionS) ORACLES (Redemann et al., 2020), Aerosol RadiatiOn and CLOuds in Southern Africa AEROCLO-sA (Formenti et al., 2019) or CLARIFY.
In both models, aging from the fresh (hydrophobic) to (hygroscopic) aged mode is quantified using an e-folding time of 6 hours according to Abel et al. (2003). This value is two times higher than ~3 h recently proposed by Vakkari et al. (2018) for the Southern African savannah. While analysis of BBA chemical composition and optical/hygroscopic properties from the recent field campaigns are ongoing, preliminary results from Ascension Island indicate that
atmospheric aging increases the ability of smoke to act as a cloud condensation nuclei and to absorb SW radiation (Zuidema et al., 2018). Finally, for each BBA tracer, log-normal dry-state aerosol size distributions and refractive indices are assumed following Mallet et al. (2017, 2019) to calculate radiative properties for «fresh» and «aged» smoke tracer. As BBA are known to be hydrophilic (Rissler et al., 2006), the dependence of the radiative properties to relative
humidity (RH) has been included for both tracers following Mallet et al. (2017, 2019).

### 2.1.4 ALADIN and RegCM experiment design

In this study, four ALADIN-Climat and two RegCM simulations have been performed (Table 1). The RegCM and ALADIN control runs (CTL) do not take BBA into account so that all aerosols are activated and interactive with

radiation (i.,e. direct and semi-direct effects for those particles are included), but biomass burning emissions are set to zero. The perturbed simulations (termed SMK) include the smoke emissions, and the direct and semi-direct radiative effects of BBA. In this study, we remind that the first indirect radiative effect of BBA is not included and the cloud droplet effective radius is fixed (10 µm). Finally, in order to test the sensitivity of DRE and SDE to the BBA absorbing
aerosols, two additional simulations, namely SMK_90 and SMK_75, have been performed with the ALADIN model

using directly fixed SSA of respectively 0.90 and 0.75 (at 550 nm) in the model. As mentioned in the introduction, the simulations using enhanced absorbing properties of BBA are motivated by recent studies showing very low SSA for aged BBA plume emitted from Central Africa (Zuidema et al., 2018, Denjean et al., 2020; Wu et al., 2020).

Global Fire Emissions Database (GFED) biomass-burning emissions (Van Marle et al., 2017) version 4 are prescribed in both models. GFED is based on estimates of burned area, active fire detections, and plant productivity derived from MODIS. Carbon emission fluxes are converted to trace gas and aerosol emissions using species-specific emission factors based on Andreae and Merlet (2001). Monthly-mean GFED emissions are used in ALADIN, while RegCM is
forced by daily mean emissions. In all experiments, the BBA emissions have been scaled up by a factor of 1.5 for BC and OC, which is a common practice in climate modelling studies for BBA (Pan et al., 2020). This factor is fairly consistent with Thornhill et al. (2018) who consider a factor of 2 in the HadGEM climate model in order to reproduce observed satellite AODs over South America. Reddington et al. (2016) indicate that multiple modeling studies have
used factors up to 6 to correctly represent observed BBA AOD from emission inventories. Johnson et al. (2016) have indicated that many studies (Marlier et al., 2013; Petrenko et al., 2012; Tosca et al., 2013) have also used emission factors higher than one.

BBAs are emitted into the first vertical level of each model, without any consideration of pyroconvective processes, as

there is no clear consensus on such processes or typical injection heights over this region. For example, Labonne et al. (2007) showed that emitted smoke plumes are generally confined to the boundary layer close to the main biomass burning source regions. Menut et al. (2018) have tested different forms of injection profiles and have shown that injection of BBAs above the boundary layer did not change significantly the impact on air quality for cities in the Gulf
of Guinea region when compared to BBAs being injected in the boundary layer. In the simulation, fire emissions from the savannah are also emitted at the lowest model level and efficiently mixed by subgrid-scale turbulence through the boundary layer. Even if the raw GFED has 3-hour intervals, the diurnal cycle of smoke emission is also not taken into account, which could impact the temporal variations of the aerosol loadings (Xu et al., 2016).
**2.2 Data**

**2.2.1 Radiation and surface temperature data**

In order to evaluate the performance of both models, we use several datasets from ground-based measurements and satellite products. The Climatic Research Unit (CRU) of the University of East Anglia provides 2m- temperature and

precipitation at a 0.5° * 0.5° resolution (Harris et al. 2013). It includes most of the land weather stations data around the world. In addition, we used the EUMETSAT CM-SAF Surface Solar Radiation Parameters (SARAH-2) which comprises five parameters related to surface solar irradiance, including surface incoming shortwave radiation (SIS). These are derived from the geostationary first generation (Meteosat-MVIRI) and second generation (Meteosat-SEVIRI)
satellite sensors. The data set covers Africa, Europe, and most of the Atlantic Ocean. Finally, we have also used the buoy observing system Pilot Research Moored Array in theTropical Atlantic (PIRATA) (Bourlès et al., 2019) for downwelling shortwave radiation in the tropical Atlantic Ocean.

**2.2.2 Cloud and Aerosol reanalysis data**

In this study, we used cloud products (liquid water path and cloud fraction) from the ERAI global atmospheric reanalysis (Dee et al., 2011) provided by the European Centre for Medium-Range Weather Forecasts (ECMWF). ERAI covers the period from 1979 onwards and has been continuously extended operationally until August 2019. The ERAI reanalysis is produced by the Integrated Forecast System (IFS), which includes the forecast model consisting of three
fully coupled components for the atmosphere, land surface and ocean waves. ERAI clouds are represented by a fully

prognostic cloud scheme in which cloud related processes are treated in a unified way; i.e. they are physically realistic and consistent with the rest of the model. Clouds are defined by the horizontal coverage of the grid box by cloud and the mass mixing ratio of total cloud condensate, along with the constraint that cloud air is saturated with regard to liquid water and ice. ERAI in general has been used in many climate studies in the past, including cloud studies (e.g. Jiang et al., 2011).

Two different reanalysis products are used to evaluate aerosols. The European Centre for Medium-Range Weather Forecasts (ECMWF) reanalysis of global atmospheric composition includes five main aerosol species. In this work, we use the recent Copernicus Atmosphere Monitoring Service (CAMS)-RA aerosol reanalysis (Inness et al., 2019) for the total AOD. In addition, we use Modern-Era Retrospective analysis for Research and Applications (MERRA)-2, generated with version 5.2.0 of the Goddard Earth Observing System atmospheric model and data assimilation system (Randles et al., 2017). We rely on the AOD for the different species at $0.5° \times 0.625°$ spatial resolution. In addition, and more specifically for the absorbing properties, we have used the recent MACv2 aerosol climatology in its second version (Kinne et al., 2019), which provides monthly global fields of optical properties at $1° \times 1°$ spatial resolution, derived from a combination of observations (notably from the AERONET network) and model outputs. The aerosol climatology is the merging of monthly statistics of aerosol optical properties with a central reference year for 2005 conditions.

### 2.2.3 Cloud and Aerosol satellite data

Spatio-temporally highly ($0.05° \times 0.05°$) resolved geostationary satellite observations are taken here from the CLoud property dAtAset based on SEVIRI edition 2 (CLAAS-2; Benas et al., 2017). The CLAAS-2 dataset is based on measurements of the Spinning Enhanced Visible and Infrared Imager (SEVIRI) and was generated and released by the EUMETSAT Satellite Application Facility on Climate Monitoring (CM SAF). CLAAS-2 includes a variety of cloud properties, including LWP, cloud optical depth and effective radius. The CLAAS-2 level 2 data are instantaneous data on native SEVIRI resolution with a temporal resolution of 15 min. For this study, the data are projected onto a regular latitude–longitude grid using the nearest-neighbor approach. It should be noted that Sc cloud retrievals could be affected by the presence of BBA over the SEA. Recently, Seethala et al. (2018) indicated that, in the aerosol-affected months of July, August and September, SEVIRI liquid water path is biased by ∼16 %.

In addition, the cloud cover has been also documented using observations from the Cloud Aerosol Lidar with Orthogonal Polarization (CALIOP, Winker et al. 2007) lidar onboard CALIPSO. The cloud cover is computed on an instantaneous basis from the CALIPSO Vertical Feature Mask version 4.20 (Vaughan et al. 2009) which provides a cloud mask on a high resolution grid up to 8.2 km, and an intermediate resolution grid (1 km horizontally and 60 m vertically) between 8.2 and 20 km. The cloud cover is computed on an instantaneous basis for three atmospheric layers located below 3.2 km, between 3.2 and 6.5 km and above 6.5 km. Because of the long revisit times of the A-Train (~16 days), the data are accumulated at seasonal time scale.

Three above-cloud AOD (ACAOD) product are used. The first is obtained from the POLDER-3/PARASOL instrument as described by Waquet et al. (2013) and Peers et al. (2015). Briefly, this is a two-step retrieval where the first step uses the polarization radiance measurements to retrieve the scattering AOD and the aerosol size distribution in a cloudy scene. In the second step, the spectral contrast and the magnitude of the total radiances measured in the visible and SWIR are used to retrieve the absorption AOD and cloud optical depth (COD) simultaneously. Therefore, the retrieval of the aerosol properties is done with minimal assumptions and with the cloud properties corrected for the overlying aerosol absorption.

Two MODIS-based products are also used. One, the Deep Blue ACAOD data set, was described initially by Sayer et al. (2016), and updated and evaluated against ORACLES field campaign data by Sayer et al (2019). In brief, this algorithm performs a multispectral weighted least-squares fit of measured reflectance in four bands across the visible spectral region to simultaneously retrieve ACAOD and COD. Finally, the MOD06ACAERO products are also used, which take reflectance observations at six MODIS spectral channels to simultaneously retrieve ACAOD, COD and the cloud effective radius of the underlying marine boundary layer clouds (Meyer et al., 2015). The main conceptual difference between these two MODIS data sets is that the former was designed primarily to extend AOD coverage into cloudy scenes, while the latter was designed to address known regional biases in cloud property retrievals resulting from the BBA signal. In addition to these above-cloud AOD data sets, two total-column AOD data products are used: MODIS Dark Target Collection and MISR (Khan et al., 2015). While the above-cloud aerosol loading is most relevant to the SDE, these total column products are used for wider context.

## 3. Evaluation

### 3.1 Surface radiation and temperature

Shortwave surface radiation from RegCM and ALADIN (control runs) have been estimated using the PIRATA buoy observations at the station 8°E/6°S. The SARAH-2 downwelling radiation data at the PIRATA buoy has been also included in the comparison. Results are provided in the Appendix (Figure S1) indicating a relatively good agreement between ALADIN and SARAH-2 especially during the biomass-burning season. A more significant positive bias (about ~40 W m$^{-2}$) is found in ALADIN when compared to in-situ PIRATA observations. This bias in ALADIN is due to the underestimation of the cloud fraction over SEA (Figure 1). The results obtained for RegCM clearly indicate a better agreement with the PIRATA observations and a slight underestimation compared to SARAH-2. Figure S1 also highlights the large difference between the PIRATA and SARAH-2 data for the period studied. Foltz et al. (2013) indicate that aerosol deposition could affect the observed surface radiation. Concerning surface temperature, the comparison with CRU data reveals (Figure S2) a positive bias of around ~1-2 K, especially over central Africa in ALADIN for the CTL run. The bias in surface temperature is more significant (~2-4 K) over the South of the Democratic Republic of Congo and Angola. RegCM simulation shows similar bias magnitude range but different spatial patterns, ranging from ~-1/-3 K for the equatorial sub-region to +4 K for the coastal Namibian sub-region. Many factors can affect surface temperature bias such as cloudiness, precipitation or boundary layer scheme. The bias showed by these regional simulation is in the range of other RCM studies realized in the frame of CORDEX (Laprise, 2013).

### 3.2 Cloud microphysical and macrophysical properties

As the first indirect effect is not treated here, the analyses are focused mainly on LCF and LWP. The seasonal (JAS) mean of LCF is shown in Figure 1 for the two RCMs and the SEVIRI and CALIOP instruments. The analyzed period is 2004-2015. First, some important differences appear between the two satellites, especially over the Gulf of Guinea and south of 25°S, where LCF is higher in CALIOP data. Compared to models, Figure 1 indicates a significant underestimation in LCF by ALADIN during the JAS season over the main Sc region, mainly between 5-20°S and 12°E-15°W. Over this zone, RegCM simulates larger LCF (~90 %), which is in better agreement with SEVIRI and CALIOP. The regional extent of Sc is well reproduced by RCMs, with a decrease above ~5°S in agreement with SEVIRI observations. Over this region, both RCMs are able to reproduce reasonably well the LCF derived from SEVIRI, especially the decrease along the Guinean coast, but an underestimation is noted compared to CALIOP. The extent of the Sc region to the south is also well captured by ALADIN and RegCM compared to SEVIRI, but is largely underestimated compared to CALIOP, especially below 20°S. The extent of Sc to the west is limited to ~10°W by the

two models, while satellite observations indicate high values up to 15°W. More specifically, the small LCF observed by SEVIRI and CALIOP along the Namibian coast is overestimated more in RegCM compared to ALADIN. Finally, over the continent, Figure 1 indicates that both models simulate LCF higher than 40% over the Gabon. In this specific region, the simulated LCF by RegCM is found to be very consistent with satellite SEVIRI observations, while in ALADIN it is more consistent with CALIOP data.

In Figure 2a, the simulated interannual variations of the seasonal-mean (JAS) LCF are also compared to SEVIRI and CALIOP observations, as well as ERAI reanalyses, over the Sc representative geographical box (10-20°S / 0-10°E) defined by Klein and Hartmann (1993) over the Atlantic. As mentioned previously, ALADIN underestimates LCF with a mean value of 63 % for the JAS season (Figure 2a) compared to SEVIRI (77%) and ERAI (75%) and CALIOP data (88%). This lack of LCF in ALADIN is consistent with the cloud biases found in its global counterpart (ARPEGE-Climat, Roehrig et al. 2020). Brient et al. (2019) attributed these biases to issues with the prescribed subgrid-scale distributions of water and temperature in the cloud parameterization and with and overestimated drying induced by the cloud-top entrainment parameterization. Concerning RegCM, the comparison indicates that the LCF is slightly overestimated during the JAS season compared to SEVIRI and ERAI, but a good agreement is obtained with CALIOP data. Since LCF does not give any indication of simulated cloud thickness which is important for radiative feedbacks, the simulated LWP is analyzed in Figure 2b. For this variable, only ERAI and SEVIRI have been considered. The results generally indicate that the two models are able to simulate consistent values compared to the observations. For ALADIN, the mean value (0.064 kg m$^{-2}$ for the CTL simulation) obtained for the 2000-2015 period generally falls within the spread of ERAI and SEVIRI LWP (0.06-0.07 kg m$^{-2}$). Figure 2b indicates that RegCM slightly overestimates LWP with a mean value of 0.08 kg m$^{-2}$. These results indicate that even though the models exhibit some important bias in LCF, which is known to be a critical unresolved problem in the global modeling communities (Nam et al., 2012), the LWP is reasonably simulated by both models. Nevertheless, the model differences and biases discussed above should be kept in mind for further analysis of the DRE of smoke exerted at TOA, especially over the main Sc region ( 10-20°S / 0-10°E).

### 3.3 Aerosol optical properties

### 3.3.1 Total column AOD

The simulated seasonal (JAS) mean AOD (at 550 nm) are reported in Figure 3 (2008-2015 period), along with the CAMS-RA and MERRA-2 reanalyses, and the MODIS Dark Target (AQUA/TERRA) and MISR satellite AOD products. Concerning the satellite data (MODIS-Terra, MODIS-Aqua, MISR), comparisons indicate important differences, both over the ocean and the continent. In particular, large differences are found between MODIS and MISR AOD retrievals with lower values associated with MISR at the regional scale. The latter is in a better agreement with the two RCMs, especially over the ocean. The difference obtained in this study between the two sensor's are in line with the recent results obtained by Sogacheva et al. (2020) over SEA. For the current MISR standard product, this study indicates that AOD is systematically underestimated for AOD > ∼0.5, largely due to treatment of the surface boundary condition at high AOD (Kahn et al., 2010). As mentioned by Mallet et al. (2019), some of the land–ocean contrast in the satellite data comes from different factors, such as the over-land and over-water algorithms, which are different and may present different biases. The second is that cloud fraction is also significantly higher over the water than over the land, meaning that typically more days of data contribute to the monthly mean over land than over water.

The magnitude of the simulated AOD is quite consistent among the two models over the ocean, but diverges over the continent where AOD simulated by ALADIN is larger, especially over the eastern part of Congo. In this region, the

difference in AOD between the two models is around ~0.2-0.3. Numerous reasons could explain these differences including the temporal frequency of the emissions (monthly vs daily) used to force the model, vertical and horizontal transport processes, optical properties (mass extinction efficiencies) such as the effect of relative humidity and wet

removal processes in connection with location and amplitude of the precipitation. Another likely contributing factor is sampling incompleteness of the satellite products, particularly over the parts of the region with high cloud cover (e.g. Figure 2 of Sayer et al., 2019). Over the ocean, the two regional models are in relatively good agreement, with AOD values of ~0.6-0.7 near the Angola/Gabon coast which decreases to ~0.4-0.5 near 0°. Figure 3 also shows higher AOD
north of the Equator in RegCM, possibly due to the fact that the simulation domain extends further north and accounts for northern hemisphere aerosol sources. In addition, RegCM and ALADIN are found to be consistent with the reanalysis data, especially with MERRA-2 AOD even if the AOD is weaker over Eastern Congo, as is the case for RegCM. Larger differences are observed between RegCM and CAMS data for the same region, while a better
agreement is found with ALADIN. The maxima of AOD is also well reproduced by ALADIN as compared to CAMS. Finally, the comparisons indicate that RegCM and ALADIN-Climat underestimate AOD north of Gabon and Congo. Finally, it should be mentioned that some biais in AOD could be due to the simulated relative humidity in the free troposphere. As shown by Mallet et al. (2019), a negative bias in the BBA extinction profiles is detected in ALADIN
simulations in its non-nudged version.

In addition to the regional distribution of total AOD, the seasonal cycle has also been analyzed in Figure 4. The different AOD estimates have been averaged over the box 15-25°E/5°S-15°S (referred to as box_S) located over the main biomass burning sources of Central Africa. This figure includes monthly-averaged AOD estimated by RegCM (2003-

2015), ALADIN (2000-2015), CAMS-RA (2008-2015), MERRA-2 (2008-2015), MACv2 (2005) and MODIS (2002-2017). The simulated AOD from ALADIN has been reported for the three different SSA used in the simulations and show very similar results. This figure indicates that both models are able to correctly simulate the order of magnitude of reanalyses, climatology and satellite AOD with the maxima between 0.4 and 0.7 during the biomass burning season,
where RegCM is particularly close to MERRA-2 AOD reanalyses. Yearly-averaged AOD indicate that both model estimates, namely ALADIN (0.27) and RegCM (0.25), are within the range of values reported by the different data-set (0.20-0.32). ALADIN is found to be consistent with CAMS-RA data in terms of AOD seasonal amplitude, even if a shift is apparent with stronger values at the beginning of the fire season in particular. This difference could be due to
precipitation biases in the ALADIN model or other aerosols advected at the boundary of the domain. Finally, the comparisons over the smoke source region point out a slight underestimation (~0.05) of AOD for the November to March period by both models as compared to CAMS and MODIS, that could be due to different reasons as the long-range transport (especially for ALADIN that does not include chemical forcing at the boundaries), emissions or some
bias in the precipitation (impact on the wet deposition). Despite these differences, the seasonal cycle of the total AOD is relatively well reproduced by both models. The temporal correlation, estimated with MODIS and MISR data, is higher (~0.95) in RegCM than in ALADIN (~0.80).

### 3.3.2 Total Above-Cloud AOD

Figure 5 displays the averaged values of ACAOD (550 nm) for the JAS period simulated by the two RCMs (SMK simulations), PARASOL, MODIS-DB AQUA, MODISACAERO AQUA and Terra. Due to the implication for semi-direct effects, this parameter is evaluated over the ocean box 0-10° E / 10-20° S where the Sc deck is present. The simulated ACAOD is underestimated (~ -0.1/-0.2) by the two RCMs compared to the MODIS-DB AQUA,
MODISACAERO AQUA/Terra and PARASOL data, with averaged-values (for the whole period) of 0.18, 0.22, 0.31,

0.31, 0.30 and 0.36 for ALADIN, RegCM, MODIS-DB, MODISACAERO (AQUA and Terra) and PARASOL, respectively. As both models have been shown to correctly reproduce total AOD near the biomass-burning sources (section 3.3.1), the differences in ACAOD, especially in 2008-2009, could be due to differences in the altitude of transport of BBA and cloud top (generally lower in RegCM) in the models linked to boundary layer dynamics and convection (possible smoke plume intrusion into the marine boundary layer), scavenging, and possibly an underestimation of humidity contained within the smoke plume which can affect optical properties as shown recently by Mallet al. (2019). The figure S3 indicates the BBA extinction (at 550 nm) and clearly shows an efficient transport of BBA plumes over the ocean in accordance with results obtained over SEA by Das et al. (2017). Extinction maxima are clearly localized between 1 and 4 km in both models but the base of the smoke plume is lower in RegCM. This may explain differences in the ACAOD between the two regional models as well as the altitude of the cloud top. The ORACLES models-observations intercomparison analysis also points to a lower extinction in the different models within the BBA layer (Shinozuka et al., 2020). While further analysis is needed, it is outside the scope of this work. However, the simulated negative bias in ACAOD is relevant to the DRE and SDE of smoke aerosols over SEA and is further discussed in following sections.

Nevertheless, the magnitude of the simulated ACAOD is consistent with other satellite-based studies. For example, during the JJA period and over the SEA, Kacenelenbogen et al. (2019) reported a seasonally averaged ACAOD of 0.25, close to the ALADIN and RegCM estimates. Based on monthly-mean time series of ACAOD over SEA using different instruments (SeaWiFS, MODIS TERRA/AQUA, VIIRS), Sayer et al. (2019) found typical values about ~0.3 during the biomass-burning season for the period 2000 to 2015. Essentially the same retrieval algorithm was applied to the four sensors.

### 3.3.3 Aerosol absorbing properties

As mentioned in the introduction, DRE and SDE of BBA are highly sensitive to absorbing properties of smoke. In order to evaluate these properties, we have compared (Figure 6) the monthly-mean SSA (for all aerosols over the whole atmospheric column and at 550 nm) obtained by RegCM (2003-2015) and ALADIN (2000-2015) with the recent MACv2 (year 2005) climatology over the box_S (15-25° E / 5-15° S). We recall that monthly sun-sky photometry statistics (from AERONET; Dubovik and King, 2000) were used as part of the MACv2 climatology (Kinne et al., 2019). The comparison indicates that the ALADIN SMK simulation is able to capture the seasonal cycle of SSA, especially between April and October. This simulation produces a SSA of ~0.85 during JJA which is consistent with the MAC-v2 data. A negative bias is present in September in which ALADIN underestimates SSA compared to MACv2. As expected by their construction, the two additional ALADIN simulations indicate lower (SMK_75) and higher (SMK_90) SSA compared to MACv2 data during the biomass-burning season. RegCM is also able to capture the seasonal variability of SSA during the June to October season, in spite of an overestimate of ~0.03-0.04.

Interestingly, Figure 6 also reveals that the ALADIN SSA is largely overestimated compared to MACv2 from November to March. This could be due to the fact that the ALADIN simulations do not take into account transport through the boundary of the domain. The lack of possible advection of BBA from Western Africa and/or mineral dust within the defined ALADIN domain could partly explain this overestimation. This positive bias is partially reduced in the RegCM simulations, which are performed on a larger domain. Finally, it should be noted that this range of simulated SSA by the two models is consistent with the SSA climatology reported by Eck et al. (2013) ~0.82-0.87 (550 nm) during the biomass burning season for the 1997 to 2005 period at the Mongu AERONET in Zambia.

### 4. Direct (SW) Radiative Effect of smoke aerosols

**4.1 Impact at the surface**

Figure 7a,b displays the JAS (SW) all-sky DRE of BBA exerted at the surface over Southern Africa for ALADIN (2000-2015) and RegCM (2003-2015). The results clearly indicate a significant decrease in solar radiation at the continental and oceanic surfaces due to BBA and its cloud response. In accordance with the simulated AOD (contour lines), in both RCMs the DRE of smoke particles at the surface is larger over the continent and decreases as the BBA plume dilutes during transport over the SEA. In general, the seasonally averaged DRE is -30/-40 W m$^{-2}$ near the biomass burning emission regions and reaches values of about -10 to -20 W m$^{-2}$ over the ocean in ALADIN and RegCM. Such estimates are consistent with those reported by Sakaeda et al. (2011) and Tummon et al. (2010) in this region. In addition, the simulated DRE over Central Africa is consistent with those reported recently by Allen et al. (2019) with a yearly-mean DRE of ~-20 W m$^{-2}$. As noted for AOD, the dimming effect of smoke in RegCM is higher over the Gulf of Guinea and in the SEA outflow than estimated in ALADIN. In addition and even if a good agreement is generally noted with the different studies, the overestimation of the LCF by RegCM over the SEA (section 3.2), in particular with respect to SEVIRI observations, may lead to an overestimation of the DRE by BBA in this model. The opposite effect is assumed in the results of ALADIN, which generally underestimates LCF.

The impact of DRE on surface temperature is analyzed in Figure 7c,d. Over the continent, a significant cooling of up to -1.0 to -2.0 K is calculated by both models. Such decreases in the continental surface temperature have already been documented in the literature by Sakaeda et al. (2011), Tummon et al. (2010) and more recently by Mallet et al. (2019), all showing similar changes. Surface cooling associated with the lower troposphere heating due to BBA has been shown to limit the development of the continental boundary layer (Tummon et al., 2010; Mallet et al., 2019). Figure 7c,d also indicates higher cooling over Southern Africa in ALADIN compared to RegCM in spite of relatively similar surface radiative forcing (Figure 7a,b), that could be due to the advection of colder air in ALADIN in the SMK simulation (see section 5.2). RegCM uses a slab-ocean model in which the impact of BBA on SST can be evaluated (Solmon et al., 2015). Figure 7c clearly indicates that the sea-surface solar radiation dimming by BBA impacts simulated SST which is regionally decreased over a large part of SEA (reaching 5° W). In this simulation, the SST cooling is not only due to the BBA direct effect, but also from a positive feedback of Sc clouds via semi-direct effects (see Section 5). We can also note an increase of SST in RegCM around 20°W, which is due to a decrease of the LCF (see Figure 12).

Figure 7c indicates that the largest SST changes, around -1 K to -1.5 K, are produced close to the Angola and Gabon coasts and collocated with AOD maximum in the RegCM simulation. However, the cooling signal is produced over a large part of SEA, from 15°S to 0° and from 8°E to 5°W, as the result of cloud feedbacks and dynamical adjustments. Over this large oceanic region, the decrease in SST varies between -0.5 and -0.2 K which is consistent with results obtained by Sakaeda et al. (2011) who also used a slab ocean model. The magnitude of the SST cooling is slightly lower in our study probably due to differences in low cloud feedbacks. As mentioned earlier, the overestimation of the LCF by RegCM over most of the SEA compared to SEVIRI may also lead to an overestimation of the impact of BBA on SST.

**4.2 Impact at the Top Of the Atmosphere**

As mentioned earlier, the sign of the overall BBA TOA radiative forcing over the SEA region is quite uncertain in GCM simulations (Stier et al., 2013). Figure 8 represents the JAS DRE simulated by ALADIN (2000-2015), RegCM (2003-2015) and MACv2 (2005). The results show a large negative DRE (~-10 W m$^{-2}$) at TOA over the continent with maxima over Angola, consistent in the two RCMs. These results are in-line with previous studies (Tummon et al., 2010; Mallet et al., 2019 and Sakaeda et al., 2011) that report significant negative TOA DRE over Southern Africa during the BBA

season. This signal over the continent is also consistent with that of the MACv2 climatology (Kinne et al., 2019), even if the magnitude is less than in the RegCM and ALADIN simulations.

Simulated TOA DRE show a dipole pattern over the SEA with positive DRE south of 5°S and negative DRE further north. This pattern is very similar between the two RCMs and in good agreement with the MACv2 data (Figure 8). This strong gradient is determined by the large decrease in low cloud fraction with latitude as one moves northwards from 5°S as shown in Figure 1, which strongly modifies the planetary albedo beneath BBA layers. As transported BBA plumes are not exactly co-located with Sc clouds (as shown by the AOD lines in Figure 8), absorbing BBA located south (north) of 5°S induce large positive (negative) DRE at TOA. In spite of the non-negligible LCF simulated over the Gulf of Guinea, the simulated cloud optical depth does not reach the critical value which would allow the BBA to switch to a positive DRE at TOA. These results clearly highlight a complex regional pattern, different than reported in the AeroCom exercise (Stier et al., 2013), which shows a more uniform (either positive or negative) DRE over SEA simulated by the different GCMs, except for CAM3, OsloCTM2 and HadGEM2-ES. More recently, Zou et al. (2020) indicate an averaged DRE (at TOA) over SEA in a present day condition very consistent (see Figure 3a of Zou et al., 2020) with the results obtained by ALADIN-Climat and RegCM.

Over SEA, simulated JAS DRE at TOA reaches a maximum of ~+5 W m$^{-2}$ for both ALADIN-Climat and RegCM. This is consistent with recent estimates proposed by Kacenelenbogen et al. (2019), who reported (using a combination of A-Train satellite sensors) seasonal-mean values of ~+2.5-3 W m$^{-2}$ for JJA and SON over SEA, including part of the Gulf of Guinea. However, the spatial extent of the positive DRE is larger in RegCM over SEA due to a larger cloud cover and thickness as well as a larger ACAOD compared to ALADIN. Differences appear notably over the Namibian coast where the sign of the forcing is opposite between the two models, which is directly associated with the large and overestimated LCF simulated by RegCM over this region. As expected, RegCM simulates larger negative DRE at TOA over the Gulf of Guinea due to larger AOD over this specific region. In continental regions, Figure 8 reveals a larger positive forcing in ALADIN over Gabon, which is certainly due to the larger LCF (see Figure 1). We argue that this positive DRE is likely to be realistic due to the co-location of BBA and persistent low level clouds over the Gabon during JAS (Philippon et al., 2019).

In spite of some regional differences in the amplitude, the two model simulations clearly highlight a remarkable gradient in the DRE of BBA. The approach of using two different independent RCMs reinforces the robustness of this original result. In addition, and although the amplitude of the DRE differs, this gradient over SEA is also clearly observed in MACv2 indicating maxima of about ~+2-3 W m$^{-2}$ over SEA and negative (-2/-3 W m$^{-2}$) over the Gulf of Guinea, as shown in Figure 8. As smoke SSA is found to be similar between the two RCMs and the MACv2 climatology (see Figure 6), the observed differences in the magnitude of DRE over SEA could be due to variances in LCF as well as ACAOD. It should also be noted that the positive DRE simulated by ALADIN and RegCM over Gabon is detected in the reanalysis data as well. As mentioned earlier and although these results appear robust compared to recent reanalyses and literature in terms of amplitude, these DRE estimates at TOA remain marred by the problem of quantifying the LCF over this region (see Section 3.2), which is inherent in climate models.

## 5. Semi-Direct Radiative Effect

### 5.1 Impact on SW heating rate and air temperature

The SDE, which represents the modifications of the cloud properties and atmospheric dynamics due to absorption of SW radiation by BBA, has been estimated based on twin simulations, one including the impact of BBAs (SMK) and the other one for which BBA emissions are set to 0 (CTL, see Section 2.1.4). The SW radiative heating due to BBA

absorption and potential feedbacks is shown in Figure 9, which displays longitude-height cross sections at two latitudes (6 and 12°S) averaged over JAS (2000-2015 for ALADIN and 2003-2015 for RegCM). The cross-sections show the differences between the SMK and CTL simulations. The results suggest that SW heating due to smoke is between +0.5 and +1.5 K by day, with higher values at 6°S compared to 12°S. The maximum of heating is located near the biomass-burning sources and decreases during the transport over the SEA to reach values around ~+0.5 K by day at ~10°W in both models. For the two RCMs, aerosol induced solar heating occurs mostly between the surface and 5 km above the surface over the continent, and between 1 to 4 km over SEA in agreement with the vertical profiles of extinction (at 550 nm, see Figure S3). Figure 9 shows that most of the additional SW heating occurs mainly above 1 km. The RegCM aerosol heating is larger than ALADIN at both latitudes, despite the fact that RegCM SSA is higher (less absorbing BBA) in RegCM (see Figure 6). This difference observed at 6° and 12°S could be due to the fact that there are more low clouds in the RegCM simulation that reinforce solar absorption within the smoke plumes. Over the continent and at both latitudes, higher solar heating in RegCM is linked to higher AODs over the source regions, especially near the coast as shown in Figure 3. This can compensate the lesser absorbing efficiency of BBAs in RegCM as compared to ALADIN. In addition, Figure 9 shows a significant heating rate increase within the Sc clouds layer for the RegCM simulation. Further discussions on this issue are detailed in Section 5.3.

The simulated SW heating rates are within the range of values reported by different studies such as Tummon et al. (2010), Gordon et al. (2018), Adebiyi et al. (2015) and Wilcox (2010). These studies have indicated additional SW heating due to smoke of 1.00 (JJAS period), +0.34 (5 days of simulations), +1.20 (for fine AOD > 0.2) and +1.50 K day$^{-1}$, respectively. In addition, Keil and Haywood (2003) estimated a SW heating rate of 1.80 K day$^{-1}$ near the coast using a radiative transfer model and observations during SAFARI-2000.

Changes in the 3D air temperature (SMK minus CTL simulations) field due to BBA are shown in Figure 10 for the same latitudes as previously used for SW heating. For the two transects, a generally good agreement is found between the two RCMs. Over the continent in both models, smoke particles are responsible for a significant decrease in air temperature between the surface and ~3-4 km height, with a higher vertical extent of cooling in ALADIN. The cooling at the surface is also more pronounced in ALADIN-Climat (~-1 K) compared to RegCM (~-0.5 K). In both models above the continent, the simulated cooling between the surface and 3-4 km height is accompanied by a general heating of the mid-troposphere (between 4 and 6 km). As noted for the smoke cooling effect, the induced heating is more significant in the ALADIN simulation at these altitudes, which can be due to a number of factors including the response of convection and dynamics to the aerosol perturbation. A detailed analysis of the change in the energy budget over the continental area is beyond the scope of the present study, but is planned in the future.

More interestingly, Figure 10 clearly highlights differences in the models response to air temperature near the surface close to the continent-ocean transition. For the two transects, the simulations differ where RegCM indicates a cooling (of about ~0.5-1 K) near the surface, which is not simulated by ALADIN. As mentioned previously, this difference is certainly due to the ocean-atmosphere coupling in RegCM that takes into account, in particular, the double impact of the BBA sea-surface forcing as well as the increase in liquid water content of Sc (part 5.3) on SST. This explains the difference in the air temperature changes obtained between RegCM and ALADIN close to the transition continent-ocean zone.

Over the Atlantic Ocean (Figure 10), the simulated air temperature response is more complex. Air temperature generally increases by 0.5-1 K between 2 and 4 km, where the core of smoke plumes are transported. At 6°S, changes in the air temperature are found in ALADIN compared to RegCM, contrary to what is observed for the heating rate (Figure 9).

ALADIN simulates an increase in air temperature (between 2 and 4 km) of about ~0.5-0.8 K, larger than RegCM (~0.2-0.5 K). In addition, Figure 10 shows that the impact of smoke aerosols on air temperature is larger at 12°S than 6°S, while the effect is opposite for the SW heating. Air temperature anomaly is not only determined by aerosol SW radiative heating, but also results from additional feedbacks including lower tropospheric dynamics and cloud adjustment
modifying the energy budget. As an example, over the continent the increase of air temperature between 5 and 7 km (at both latitudes) above the surface could be due to increase of the vertical ascent (see Figure 12) of (hot) air masses. A specific study investigating changes in all the terms of the air temperature tendency would allow to quantify the different impacts. The 2 to 4 km temperature changes obtained in this study are in a good agreement with values
published by Sakaeda et al. (2011) (+0.5 K), Allen and Sherwood (2010) (+0.5-1 K at 700 hPa and for the JJA period) and more recently by Gordon et al. (2018) (+0.4 K).

Under the smoke plume, RegCM and ALADIN both show a similar temperature response in a very tight layer, located between 1 and 2 km, which is cooled by ~-0.5/1 K (up to about 10°W). This cooling could result from the additional

scattering of solar radiation by the smoke plume located above, but is likely to also be driven by additional LW cooling at the top of cloud layer due to the increase of Sc water content as a results of SDE (see Figure S4 in Appendix). Finally, temperature changes in the marine boundary layer (MBL, surface to ~1km) are quite different between the two RCMs, especially at 6°S. The MBL is homogeneously heated by about ~+0.5 K in ALADIN whereas RegCM exhibits a
cooling, especially near the coast. As mentioned previously, this is linked to the slab-ocean parmeterization and SST cooling propagating to the MBL via turbulence in the case of RegCM. For ALADIN, heating of the MBL could be due to the LW trapping due to the increase of LWP and LCF at 6°S notably.

**5.2 Impact on the sea-level surface pressure and circulation**

For the first time to our knowledge, we have investigated in this work the SDE of BBA on the lower tropospheric dynamics in Central Africa and SEA. Figure 11a,b displays changes in sea-level surface pressure (SLP) between the SMK and the CTL simulations for the two RCMs and for the JAS period. A dipole pattern showing a cyclonic anomaly over SEA and an anticyclonic anomaly over Congo/Angola is obtained for both models, despite geographical
differences over SEA. Over the continent, the regional patterns of SLP changes are quite consistent, even if the maxima of the positive anomaly over Angola is higher in ALADIN (+50 Pa) compared to RegCM (+40 Pa). The anticyclonic anomaly is related to changes in the lower tropospheric radiative budget which is induced by BBA. As reported for the air temperature changes, lower troposphere cooling (associated with heating above 4 km) generally increases the
stratification over the continent. This results in a more stable atmosphere and a decrease in vertical velocity between the surface and 4-5 km (Figure 12). This impact of BBA over the continent is consistent with results obtained by Sakaeda et al. (2011) and Allen and Sherwood (2010). The latter indicates an increase in lower tropospheric dry static stability over Central Africa during the JJA period based on the NCAR CAM3 GCM coupled model. More recently, Allen et al.
(2019) have also reported a general increase in LCF and lower tropospheric stability (estimated between 700 hPa and the surface) over Central Africa using three different GCMs.

Over SEA, the two vertical velocity transects (Figure 12) indicate that the subsidence is reduced, with maxima located between 2 and 4 km, which is consistent with Sakaeda et al. (2011) findings. Adebiyi et al. (2015) also indicate that

ERAI subsidence is less when there is more smoke aerosol present. The decrease of the tropospheric stability in both RCMs is likely due to the anomalous radiative heating in the aerosol layer (see Figure 9) that enhances buoyancy. This is associated with a cyclonic circulation anomaly over most of the SEA and a low pressure anomaly of ~30-40 Pa at the sea surface (Figure 11a,b). This anomaly creates a change in the Sc cloud tops in the SMK simulation with, in

particular, an increase of about ~30hPa (Figure S5). Over SEA, the difference between the two models is more pronounced than over the continent and the negative anomaly SLP is located further west and south in RegCM, and found to be lower (-10/-20 Pa), than in ALADIN (-30 hPa). The decrease of SST in RegCM results in a local enhancement of stability, quite similar to those produced over the continent, especially near the coast where the AOD is high. Some differences appear also near the Angola coast, where RegCM simulations indicate an increase in the SLP (~20 Pa), which is not simulated by ALADIN. As mentioned previously, the difference is due to a significant decrease in SST (~-1.5 K) in RegCM due to the BBA dimming effect near the Angola coast (see Figure 7a). Over this specific region, the results obtained by RegCM are in agreement with those of Sakaeda et al. (2011) who report an increase of the lower tropospheric stability over a large part of SEA due to BBA direct and semi-direct effects.

This SLP anomaly creates some changes in the surface wind speed and direction as shown by the Figure 11c,d. Over SEA in the ALADIN model, the negative cyclonic anomaly generates more westerly winds over the Gulf of Guinea (~0.4-0.5 m s$^{-1}$) and increases the north wind along the coasts of Angola and Congo by ~0.3 m s$^{-1}$. In the RegCM model due to the position of the anomaly, the changes in the wind fields are slightly different and an intensification of northwest winds (by ~0.6 m s$^{-1}$) between 0° and 10°S is simulated. Moreover, the increase in northerly winds near the coast of Angola detected in ALADIN is more pronounced in RegCM and reaches values of ~0.6-0.7 m s$^{-1}$.

**5.3 Impacts on Sc properties**

In addition to the SDE of BBA on SLP and the atmospheric surface circulation, the impacts on Sc properties have been analyzed and are shown in Figure 11e,f and 12. Over the continent, both RCMs simulate an increase in LCF and LWP associated with enhanced lower tropospheric stability as discussed previously. In ALADIN, the increase in LCF maxima (~7%) are located over Gabon and Eastern Congo. More generally over Congo, the LCF is increased by about 2-5%. RegCM also produces higher LCF induced by BBA, but the impact is generally lower ~1-2%. These results are similar to those recently found by Allen et al. (2019) who report a 5% increase in LCF induced by fine aerosols using different GCMs (CAM4, CAM5 and GFDL). However, Sakaeda et al. (2011) report a decrease of the continental LCF. Reasons for this discrepancy would require a more detailed model intercomparison. Figure 11 indicates also a general increase of LCF along the Gulf of Guinea coast for the two RCMs, which is consistent with the recent work of Deetz et al. (2018), who indicate a negative feedback of the stratus-to-cumulus transition with increased aerosols during the DACCIWA experiment.

Over SEA, the LCF response pattern is quite different between the RCMs. While marked regional heterogeneous changes appear in ALADIN, a more uniform increase of LCF is obtained in RegCM. Nevertheless, Figure 11c,d indicates that the sign of the LCF changes is consistent between the two RCM over the main Sc zone (0-10°E/10-20°S), indicating a moderate increase of about ~2-5%. This increase is also shown in Figure 2a where there is a moderate increase in LCF (~2-4%) in the SMK ALADIN and RegCM simulations compared to the CTL runs over box_O. Concerning the microphysical properties of Sc, Figure 2b indicates similar results for the LWP (over the box_O) with an increase of about ~6-7% for ALADIN and ~10% for RegCM. This is also clearly indicated in Figure 12 for the transect at 12°S, showing an increase of the cloud liquid water content (by ~+0.01/0.04 g kg$^{-1}$) over the ocean in both models. The general increase in LCF and LWP over the Sc region is certainly due to an enhanced buoyancy above the MBL due to BBA SW heating, limiting the entrainment of dry air from the free troposphere within marine boundary layer, as proposed by Wilcox et al. (2010) and Johnson et al. (2004). This impact can be clearly seen in Figure 12, which shows a reduced large scale subsidence over the ocean for the two transects at 6 and 12°S, as mentioned earlier.

North of 10°S, ALADIN simulates a decrease in LCF contrary to RegCM. The negative impact obtained in ALADIN
could be due to the decrease of latent heat fluxes (see Figure S6 in Appendix) in the SMK simulation over this region,
which limits humidity input in the MBL. The difference between the two models is also clearly shown in Figure 12 for
the transect at 6°S, where a decrease in the liquid water content (of about ~-0.01 g kg$^{-1}$) appears in ALADIN over the
ocean. At the same latitude, RegCM indicates on the contrary an increase in the water content of ~+0.04 g kg$^{-1}$.

Compared to recent literature, the decrease in LCF simulated by ALADIN is found to be consistent with recent findings
of Zhang and Zuidema (2019) who report a low cloud cover decrease with enhanced smoke loadings at Ascension
Island (8°S, 14.5°W). In addition, the north-south gradient in the LCF changes obtained in ALADIN is remarkably
consistent with the recent findings of Allen et al. (2019), showing similar impacts for 2 of the 3 GCMs used in their
study. For the CAM4 and GFDL models, the radiative impact of fine mode aerosols leads to a regional pattern of
increased/decreased LCF over SEA, similar to that found in ALADIN-Climat. On the contrary, these changes in LCF
differ from Sakaeda et al. (2011) who indicate a more uniform positive impact (increase of LCF) over SEA in
agreement with the RegCM simulations. At this stage, it seems that the use of an atmosphere coupled to a slab ocean
leads to more uniform responses (positive cloud feedback over most of the SEA) compared to atmospheric models only
(using prescribed SST) such as ALADIN and Allen et al. (2019). The increase in the LCF over the ocean creates
generally a negative semi-direct effect at TOA over the SEA (Figure S7) especially for the RegCM model. The values
are comprised between -2 and -10 W m$^{-2}$, slightly higher that the mean value (-3.0 W m$^{-2}$) reported by Sakaeda et al.
(2011) over SEA at a climatic scale. For the ALADIN model, positive and negative semi-direct forcing is present over
the ocean due to different changes in the LCF. Finally, the positive semi-direct forcing over the continent is mainly
related to the response of high clouds in both models.

**6. Sensitivity of the direct and semi-direct effect to smoke absorbing properties**

In this section, the sensitivity of the different BBA impacts to smoke absorbing properties have been tested using the
ALADIN model. As mentioned earlier, two additional simulations (referred to as SMK_75 and SMK_90) were
performed for the same period (2000-2015) where the smoke SSA has been changed to 0.75 and 0.90, respectively.
Figure 13 displays the DRE of BBA exerted at TOA (in all-sky conditions) for the three different ALADIN runs. Over
the continent, as expected the results indicate an increase of the cooling effect of BBA at TOA (~-10/-15 W m$^{-2}$) for the
more scattering simulation (SMK_90). The opposite is obtained for SMK_75 in which the DRE significantly decreases
to ~-3/-6 W m$^{-2}$ over the continent. As the AOD over the continent remains constant between the different ALADIN
simulations over the main BBA sources (Figure 4), these significant changes in the TOA DRE are mainly due to the
different absorbing properties and related adjustments. For the SMK_75 simulation notably, the large DRE changes at
TOA compared to the SMK and SMK_90 simulations are also related to a increase in the LCF in SMK_75 (see Figure
S8), as well as the enhanced absorbing efficiency of BBA. Both the aerosol surface dimming effect and the tropospheric
radiative heating are enhanced in the case of SMK_75 compared to SMK and SMK_90 (not shown). This results in
additional stratification and low-level clouds over the continent in SMK_75 (Figure S8). The higher LCF increases the
planetary albedo beneath the aerosol layers, which, combined with strongly absorbing smoke, significantly decreases
the DRE of BBA at TOA over Central Africa compared to the CTL or SMK_90 runs (Figure 13). Contrarily, SMK_90 is
characterized by lower LCF resulting in more significant cooling at TOA. These results highlight the complex
feedbacks between BBA and low cloud properties modulating the DRE of smoke aerosols at TOA over Central Africa.
Over SEA, Figure 13 indicates considerable variability in the DRE at TOA among the three different simulations. As
expected, the DRE exerted at TOA by BBA over the Sc zone is greatly increased in the SMK_75 simulation compared

to the SMK or SMK_90, and reaches values of ~+5-10 W m$^{-2}$ during the JAS season. The changes are quantified in Figure S9, which shows the JAS DRE over box_O for each simulation. DRE varies from +0.94 W m$^{-2}$ for SMK_90 to +3.93 W m$^{-2}$ for SMK_75. Changes in the DRE at TOA are less significant when comparing the SMK and SMK_75 runs, with values of +3.21 and +3.93 W m$^{-2}$, respectively. Over the Gulf of Guinea, changes in the DRE exerted at TOA are opposite, and as expected the DRE increases in the SMK_90 simulation, when BBA scattering is enhanced. Over a darker ocean, compared to the Sc region, BBA induce a cooling effect at TOA which is enhanced for higher SSA, reaching a maximum of about -5 W m$^{-2}$. The cooling increase at TOA for higher SSA could also be amplified by the moderate decrease in LCF found in the SMK_90 simulation, which results in a lower planetary albedo over the Gulf of Guinea (see Figure S8) and a more negative TOA forcing.

## 7. Conclusions

This modeling study presents an analysis of the DRE and SDE of absorbing BBA over Southeastern Atlantic using decadal simulations from two different regional climate models. ALADIN uses prescribed sea surface temperatures, while RegCM includes a slab-ocean model. Both RCMs struggle to represent the LCF over SEA, which is a recurring problem in climate models (Nam et al., 2012), but the integrated liquid water content is fairly well modeled. This leads to uncertainties in the estimated DRE. For the JAS season, the simulated ALADIN and RegCM AODs are found to be consistent with the MERRA-2 and CAMS-RA reanalyses, contrary to the simulated ACAOD which is slightly underestimated compared to satellite data for the two models. The DRE exerted at the surface by BBA is significant in both models and varies regionally between -10 and -50 W m$^{-2}$, having significant impacts on continental and ocean surface temperatures. At TOA, the simulations indicate a remarkable SW DRE regional contrast in all-sky conditions for both models, in agreement with the recent MACv2 aerosol climatology. The TOA DRE is positive and around ~+3-6 W m$^{-2}$ over the Sc region. This important dipole over SEA is created by the transport of absorbing BBA both over low and high LCFs.

ALADIN and RegCM simulations indicate that BBA are responsible of an additional SW radiative heating of ~+0.5-1 K by day over SEA during JAS, with maxima located at an altitude between 2 and 4 km. The changes in the air temperature profile are shown to inhibit subsidence over SEA, creating a cyclonic anomaly at the sea-level pressure. The opposite effect (anticyclonic anomaly) is simulated over the continent by both models due to the increase in lower troposphere stability. Regarding the SDE of BBA on low-clouds, both models moderately increase LCF by about ~5% over the Sc region but their impact differ over the Gulf of Guinea. These differences in SDE are likely due to the ocean-atmosphere coupling in RegCM only where changes in SSTs increase lower troposphere stability and LCF over SEA.

Two additional ALADIN simulations have been performed with different SSAs (0.75 and 0.90 at 550 nm) and indicate that the DRE and SDE are sensitive to the absorbing properties of smoke. Over Central Africa, feedbacks between BBA and low cloud properties, and so the surface albedo, contribute, in addition to the intrinsic absorbing properties of smoke, to modulate the DRE at TOA. Over the Sc region, the positive DRE is significantly increased for lower SSA simulations with moderate SDE changes on low clouds. All the identified changes induced by BBA radiative effect on latent heat fluxes, lower troposphere atmospheric circulation and SST could possibly impact regional precipitation and dynamics (Western African Monsoon system) and need to be investigated in the future.

**Acknowkedgments**

This work was supported by the French National Research Agency under grant agreement n° ANR-15-CE01-0014-01, the French national program LEFE/INSU, the Programme national de Télédetection Spatiale (PNTS, http://www.insu.cnrs.fr/pnts), grant n° PNTS-2016-14, the French National Agency for Space Studies (CNES), and the South African National Research Foundation (NRF) under grant UID 105958. The research leading to these results has
received funding from the European Union's 7th Framework Programme (FP7/2014-2018) under EUFAR2 contract n°312609″. This work was granted access to the HPC resources of CALMIP supercomputing center under the allocation 2019- p19062.

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

|  | ALADIN | RegCM |
|---|---|---|
| Horizontal resolution | 12 km | 80 km |
| Number of vertical level | 91 | 42 |
| Emissions | GFED (monthly) Van Marle et al. (2017) | GFED (daily) Van Marle et al. (2017) |
| Scale factor | 1.5 for OC and BC | 1.5 for OC and BC |
| Aerosols types | Mineral dust, primary sea spray, biomass burning, anthropogenic (BC, OC, SO4) | Mineral dust, primary sea spray, biomass burning, anthropogenic (BC, OC, SO4) |
| Mixing assuption (optical calculations) | External | External |
| BBA SSA for sensitivity experiments (at 550 nm) | 0.75 (SMK_75) & 0.90 (SMK_90) | # |
| Aerosol Boundary Conditions | No | Yes (CAMS) |
| Ocean-Atmosphere coupling | No (prescribed SST) | Yes Slab-ocean model |
| Radiative Transfer Scheme | FMR (SW) / RRTM (LW) | RRTM (SW & LW) |
| Period of simulations | 2000-2015 | 2003-2015 |

**Table 1. RegCM and ALADIN regional climate model configurations.**







**Figures**

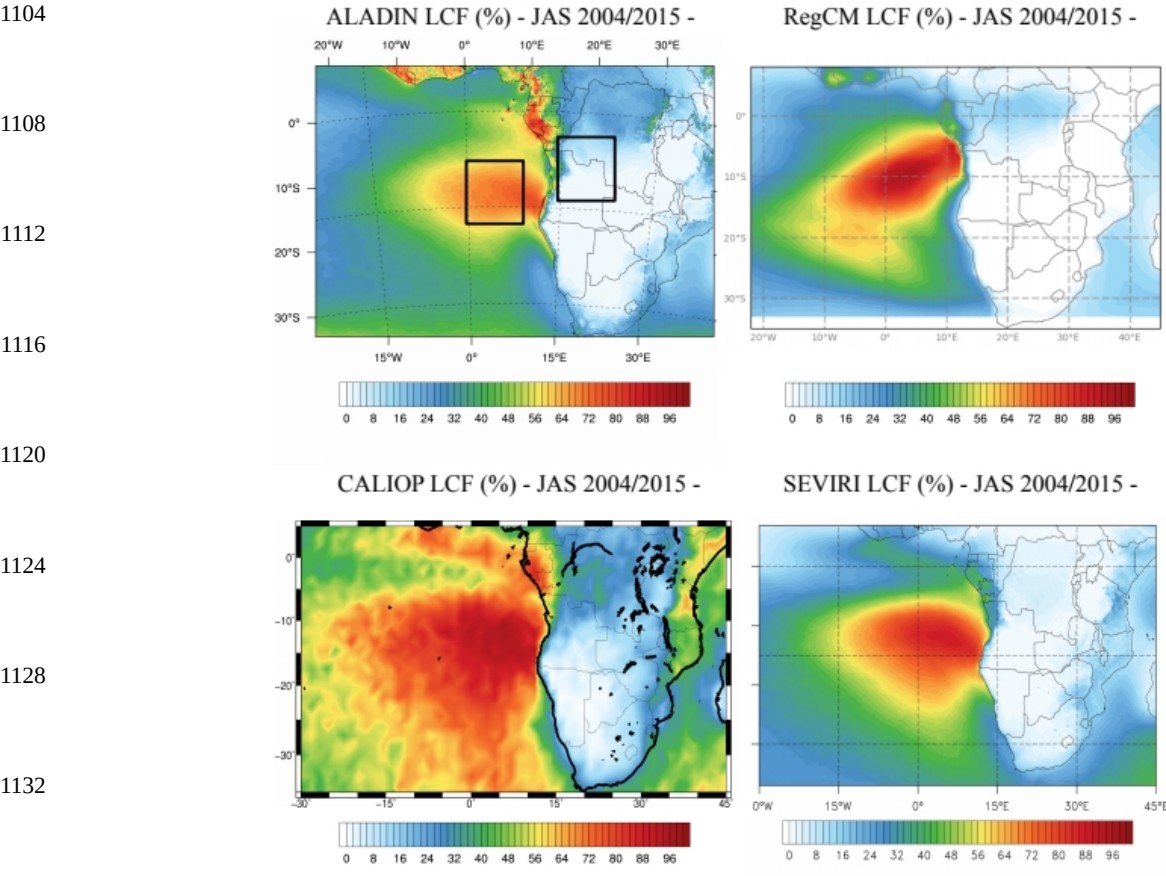


Figure 1. Seasonnal (JAS) mean of the LCF (%) simulated for the ALADIN (2004-2015), RegCM (2004-2015) models (CTL runs) and retrieved by the SEVIRI and CALIOP (2004-2015) instrument. The two different boxes (Box_0 and Box_S are indicated). The Box_0 (10-20°S / 0-10° E) has been defined by Klein and Hartmann (1993).







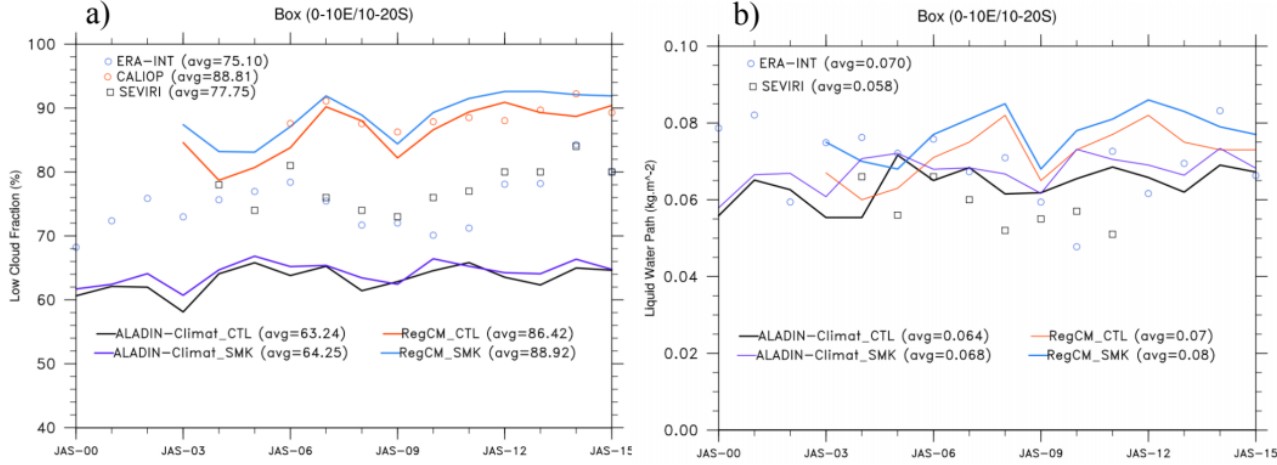

**Figure 2. a) Low Cloud Fraction (%) (left) and b) Liquid Water Path (kg m⁻²) (right) obtained by CALIOP, SEVIRI, ERA-Interim (grid-box mean) and the two regional models over the Box_0 (10-20°S / 0-10° E) defined by Klein and Hartmann (1993). CTL and SMK simulations are shown for both models.**











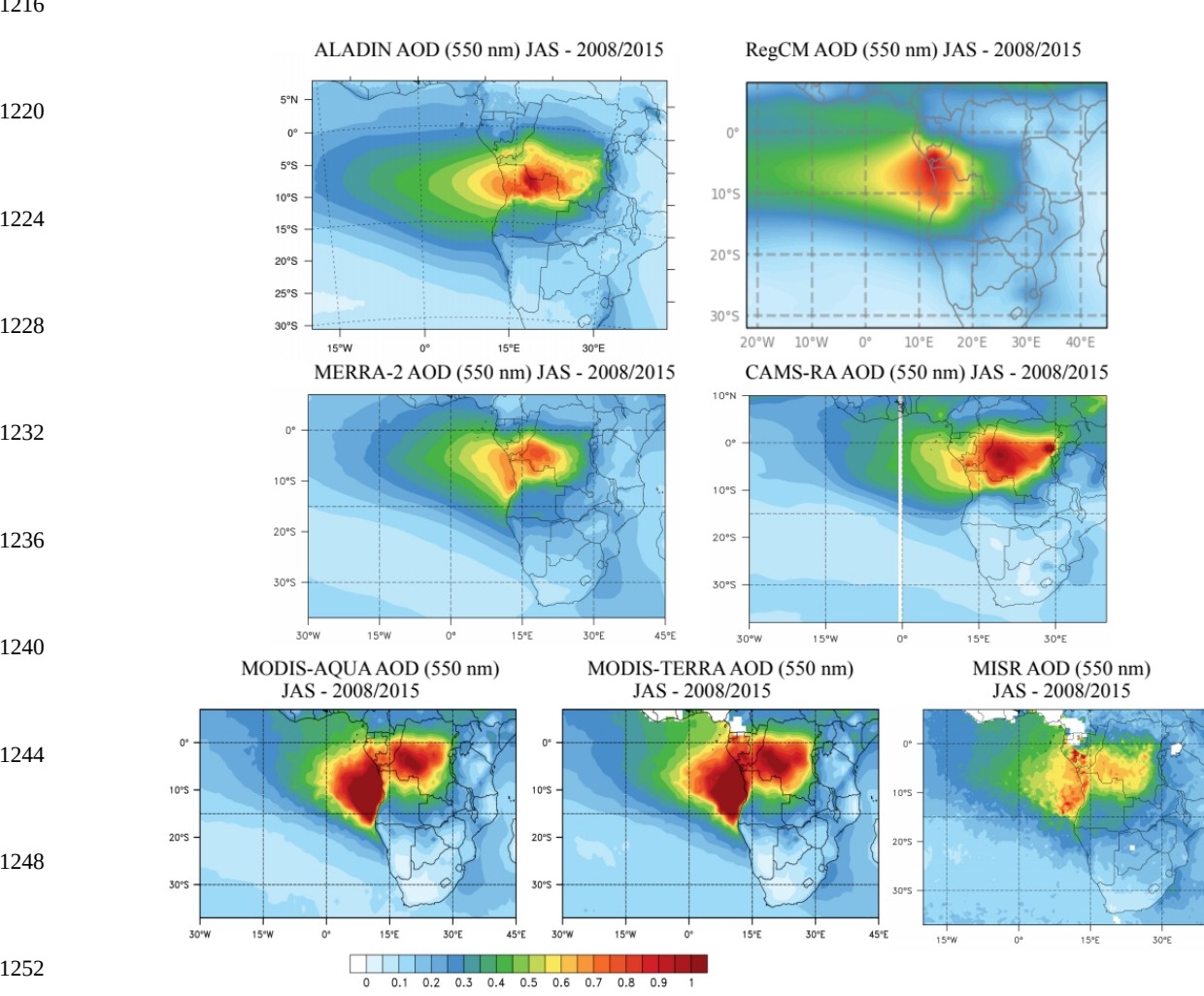










**Figure 3. Total Aerosol Optical Depth (AOD) estimated at 550 nm by the two RCMs (ALADIN and RegCM for the CTL runs), two reanalyses (CAMS-RA and MERRA-2) and two satellite products (standard MODIS and MISR AOD). The different period of observations and simulations are reported.**















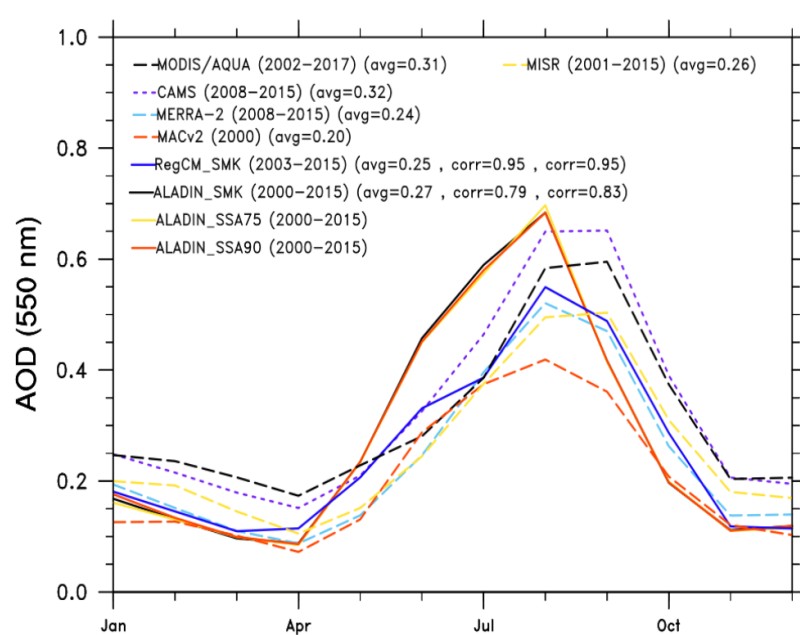



**Figure 4. Monthly-mean total AOD (550 nm) averaged over the Box_S (15-25E/5-15S) for the MODIS/AQUA (standard AOD) and MISR instruments, CAMS-RA and MERRA-2 reanalyses, ALADIN and RegCM models. For ALADIN-Climat, the CTL, SMK_75, SMK_90 simulations are reported. The different periods of the observations and simulations are indicated. The AOD temporal correlation for each models, estimated with MODIS and MISR data, are also reported.**

















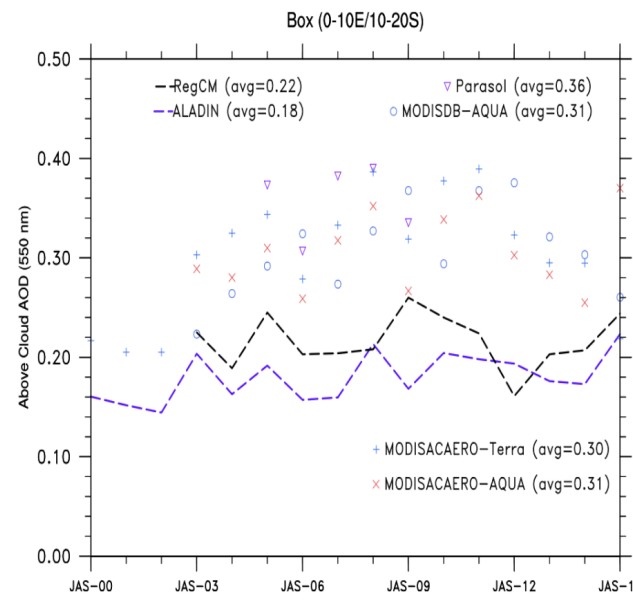


**Figure 5.** Seasonal (JAS) mean of the total ACAOD (550 nm) averaged over the box_O. RegCM (2003-2015), ALADIN (2000-2015) SMK simulations and PARASOL (2005-2009), MODIS Deep BlueAQUA (2003-2015), MODISACAERO Terra (2000-2015) and MODISACAERO AQUA (2003-2015) satellite observations are reported.














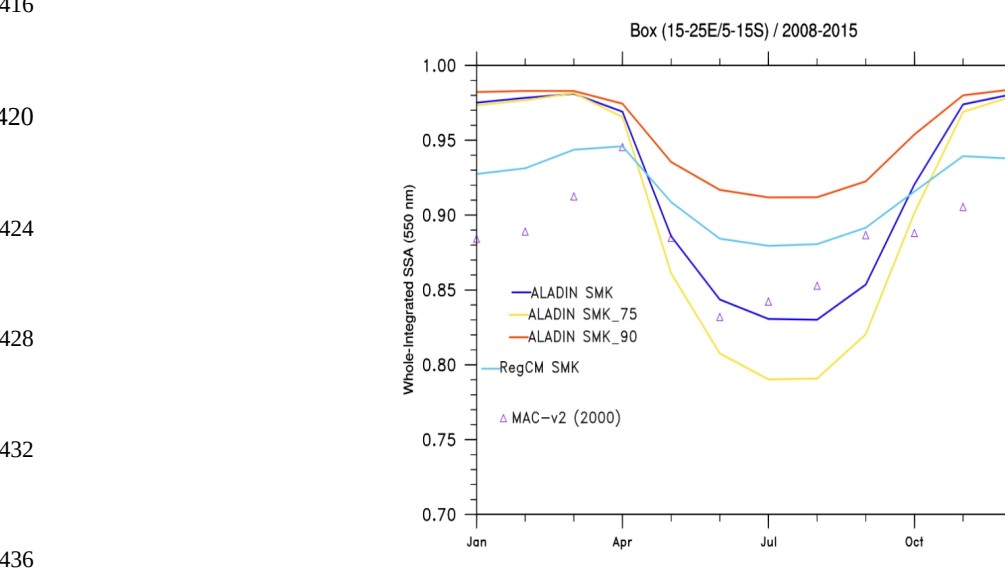

**Figure 6. Monthly-mean SSA (550 nm) averaged over the Box_S (15-25E/5S-15S) for the ALADIN (2000-2015), and RegCM (2003-2015) models and the MACv2 climatology. The CTL, SMK_75, SMK_90 ALADIN simulations are shown.**






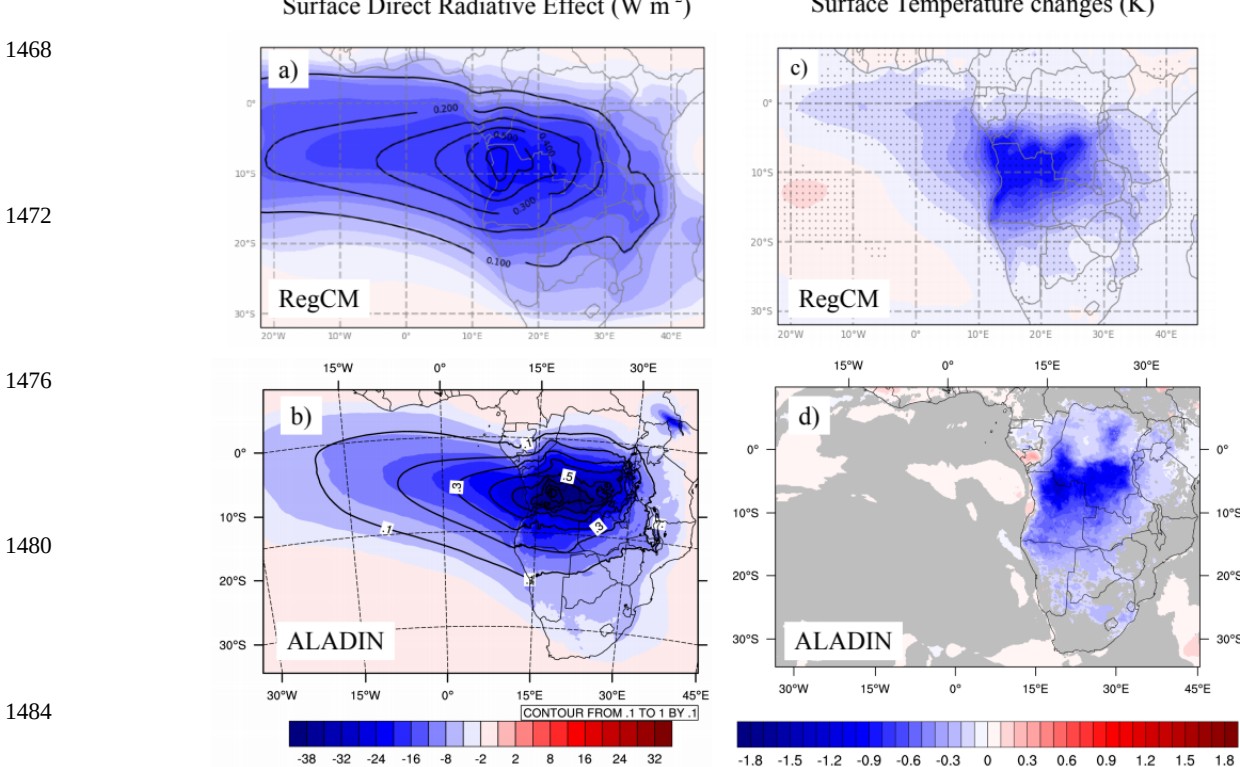

**Figure 7.** Seasonal-mean (JAS) DRE (W m$^{-2}$) exerted by BBA at the surface in the shortwave (all-sky conditions) for the ALADIN (left, down) and RegCM (left up) models. The AOD of BBA are indicated by the black lines. Seasonal-mean (JAS) changes in the surface temperature due to the BBA DRE for the ALADIN (right down) and RegCM (right up). For the surface temperature map, the grey (not dashed) areas are not statistically significant at the 0.05 level for ALADIN (RegCM).

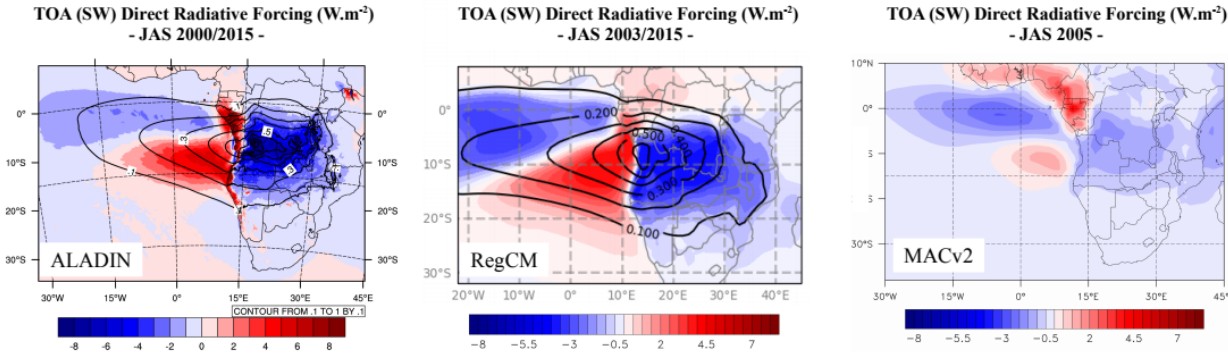

**Figure 8. Seasonal-mean (JAS) BBA DRE (W m⁻²) exerted at TOA in the shortwave (all-sky conditions) for ALADIN (left, period 2000-2015), RegCM (middle, period 2003-2015), and the MACv2 climatology (right, year 2005). The ISCCP-based cloud cover for high (<440 hPa), middle (440-680 hPa) and low (> 680 hPa) altitudes are used for the MACv2 radiative transfer calculations. The AOD of BBA are indicated by the black lines.**












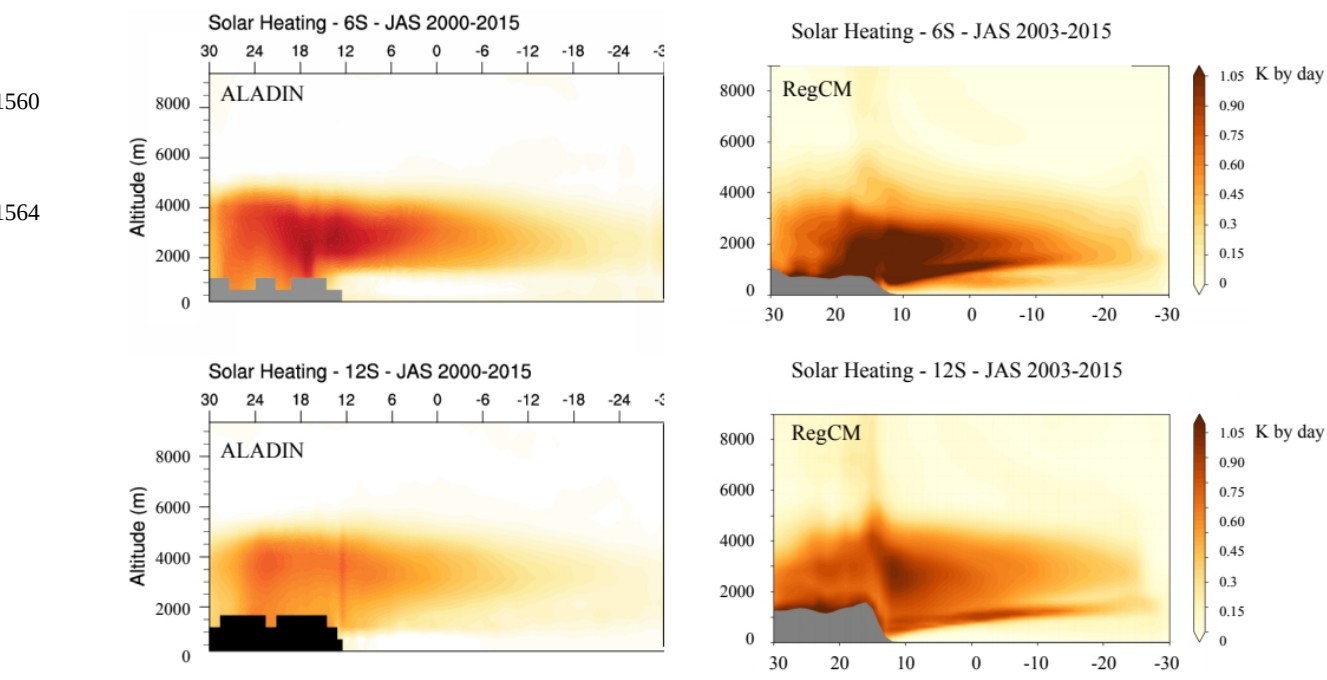

**Figure 9. Seasonal-mean (JAS) changes (SMK minus CTL simulations) in the vertical profiles of SW heating rates (K by day) due to BBA at two latitudes (6 and 12°S), for the ALADIN (left, period 2000-2015) and RegCM (right, period 2003-2015) models.**

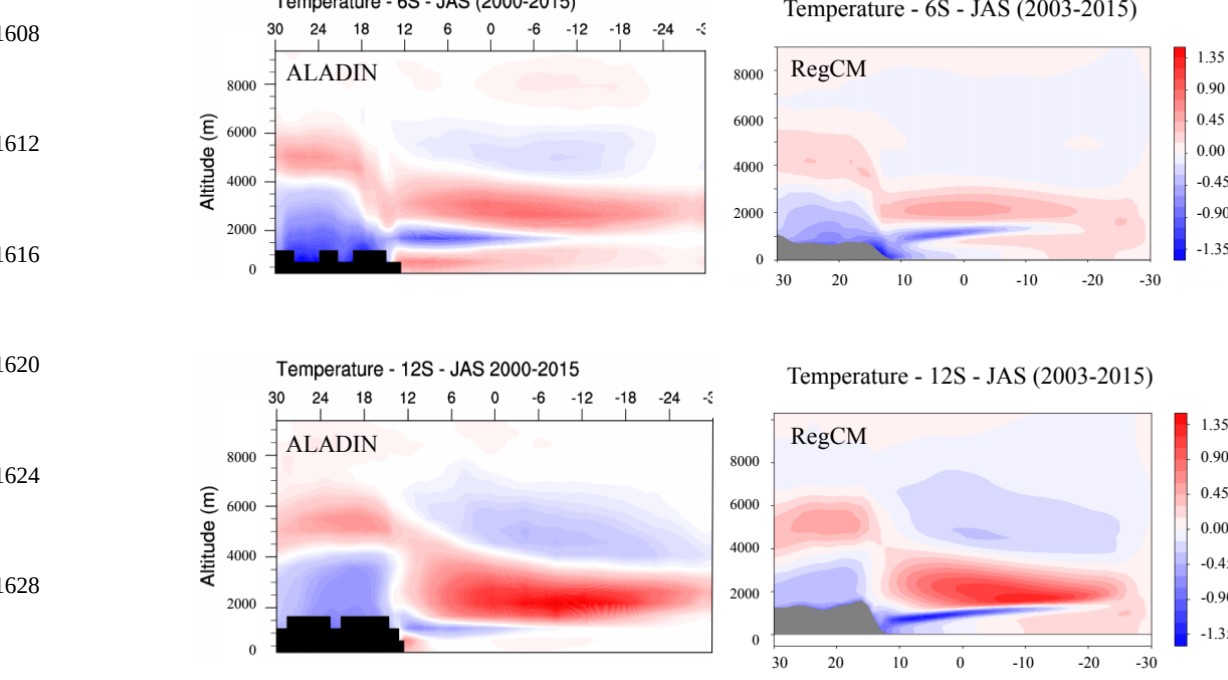

**Figure 10. Seasonal-mean (JAS) changes (SMK minus CTL simulations) in the vertical profiles of air temperature due to BBA at two latitudes (6 and 12°S), for the ALADIN (left, period 2000-2015) and RegCM (right, period 2003-2015) models.**

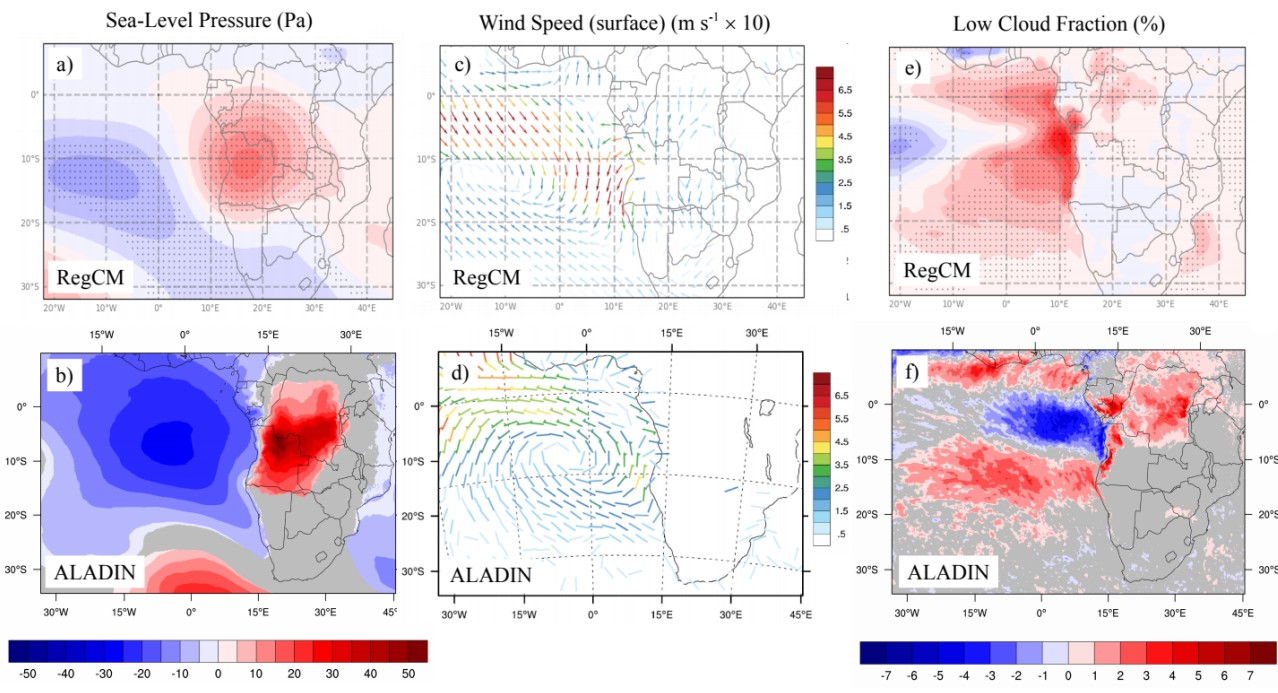


**Figure 11. Left column: seasonal-mean (JAS) changes (SMK minus CTL simulations) in the Sea-Level Pressure (SLP in Pa) for the ALADIN (left down, period 2000-2015) and RegCM (left up, period 2003-2015) models. Right column: seasonal-mean (JAS) changes in the LCF. The grey areas in ALADIN maps (not dashed in RegCM maps) are not statistically significant at**
**the 0.05 level.**








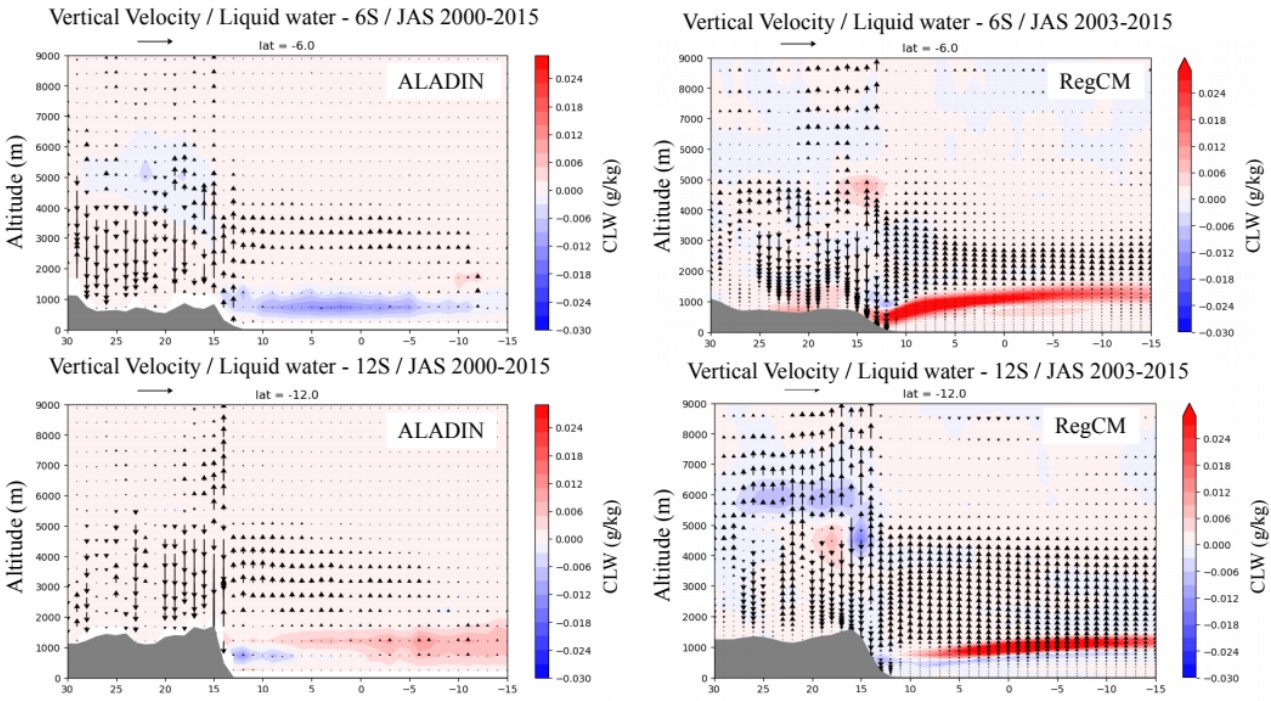

**Figure 12. Seasonal-mean (JAS) changes (SMK minus CTL simulations) in the vertical profiles of the vertical velocity (arrow) and cloud liquid water content (in g by kg) for ALADIN (left, period 2000-2015) and RegCM (right, period 2003-2015).**








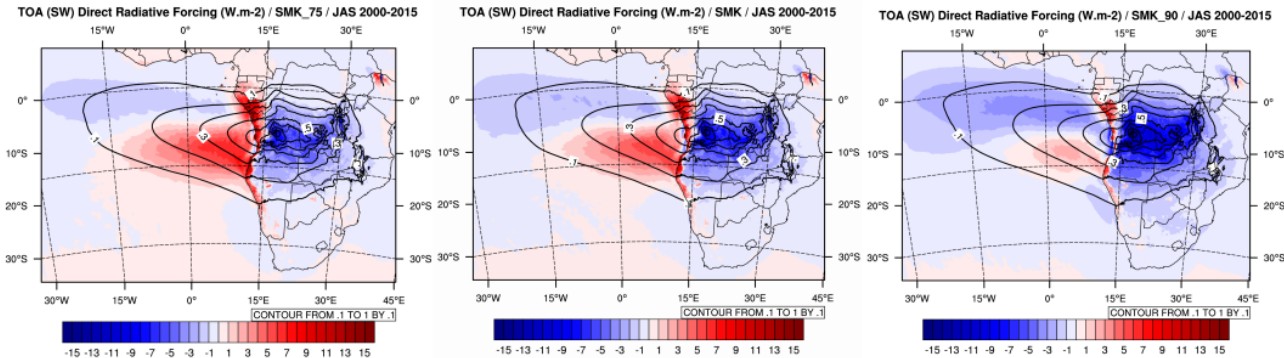

**Figure 13.** **Seasonal-mean (JAS) BBA DRE (W m⁻²) at TOA exerted in the shortwave (all-sky conditions) for the three ALADIN simulations (SSA of 0.75, left; 0.85, middle and 0.90, right) and for the period 2000-2015. The AOD of BBA are indicated by the black lines.**