# Peer review of "Direct and semi-direct radiative forcing of biomass burning aerosols over the Southeast Atlantic (SEA) and its sensitivity to absorbing properties: a regional climate modeling study."

_Atmospheric Chemistry and Physics, 2020_

## Referee Comment (RC1) · Anonymous Referee #2 · 11 Jun 2020

**Review of Mallet et al, Direct and semi-direct radiative forcing of biomass burning aerosols over the Southeast Atlantic (SEA) and its sensitivity to absorbing properties: a regional climate modeling study.**

Two atmospheric model simulations of direct and semi-direct effects of biomass burning aerosols over the south-east Atlantic ocean are compared. Simulations are performed over long enough periods to be climatically interesting. The main differences between models are that ALADIN uses prescribed SST and 12km horizontal resolution, RegCM does not and has 80km resolution. Both models use single-moment aerosol microphysics and neither represents indirect effects of aerosols on clouds. The model evaluation is excellent, with maximum use being made of the latest satellite products and detailed insights noted on the differences between various observational products.

Biomass burning aerosol causes significant shortwave heating, consistently with other studies. The models significantly underestimate cloud fraction. This underestimate means they will be interesting to compare with other models that overestimate cloud fraction (eg Unified Model, Gordon *et al* 2018), but they get LWP about right – suggesting the clouds must also be too optically thick.

I think the topic is important, the analysis is sound and the paper is very well-written. It is well suited for publication in ACP and should be highly cited. My suggestions and comments, overall, are minor.

**Scientific comments**

Can you speculate further about why ALADIN-Climat underestimates the cloud fraction? Currently saying 'cloudiness, precipitation or boundary layer scheme' is a bit vague, though I appreciate the simulations are expensive and it may not be possible to do diagnostic sensitivity studies.

L340 can you dig a bit deeper into the huge difference between MODIS and MISR AOD? Could it be due to cloud masking? Maybe reference papers where the two retrievals have been compared to AERONET?

Line 549- what about changes in inversion/cloud top height with smoke aerosol? Are there any such changes? Lines 568-580 say subsidence is reduced by smoke and tropospheric stability is decreased, but perhaps you can spell this out a bit more? Dipole patterns in Figures 12 (bottom-right) and S4 suggest cloud height changes might be occurring.

One other thing missing from the analysis is an evaluation of free-tropospheric (and boundary layer) relative humidity. Do the models replicate the observed increases in RH associated with, or coincident with, smoke layers?

Finally, I was expecting to see a discussion of the net semi-direct radiative effect of the BBA in $Wm^{-2}$. If the DRE is $(F-F_{clean})_{SMK}-(F-F_{clean})_{CTL}$, as per Ghan (2013), can the SDRE be calculated in these simulations as $F_{clean,CTL}-F_{clean,SMK}$? (not sure I got the sign right, but you get the idea). We would presumably expect RegCM to show a negative SDRE as Figure 11 shows an increase in cloud fraction (albeit perhaps not statistically significant) and Figure 12 increases in LWC, while ALADIN will have a positive SDRE in some regions.

**Text comments and suggestions**

Line 300: missing "such" in "such as"?

The text says both models are "driven" by ERA-interim reanalysis – does that mean the boundary conditions are derived from ERA-interim or is there nudging of horizontal winds?

How do the two models handle sub-grid cloud?

What does MACv2 do for the cloud fraction/water path? Or what is used for the cloud fraction in the calculation to get the DRE from MACv2 on the right of Figure 8?

Models assume external mixing of aerosols. Both represent fresh and aged smoke, but there is not a separation between BC and OC. This is interesting and complementary to other models. Both use scaled GFED emissions (could additionally cite https://www.atmos-chem-phys.net/20/969/2020/ to justify this). Which version of GFED?

Line 460 When reading this, at first I got slightly confused between decreases in cloud fraction in SMK compared to CTL (which are irrelevant here) and decreases in cloud fraction from south to north, which is what you are talking about. Maybe rephrase to "decrease in low cloud fraction with latitude as one moves northwards from 5S" or similar.

Line 495 CTRL->CTL.

Line 668 would be good to put some numbers on the TOA DRE as you do for the surface DRE.

Figure 1 would be helpful to reference Klein and Hartmann in the caption, as in Figure 2.
Figure 2 can yellow be replaced by orange for RegCM_CTL, so it doesn't fade into the background?
Figure 2b do you show grid-box-average or in-cloud LWP?
Figure 3 specify that (if I am correct) this is MODIS standard AOD, not MODIS ACAOD.
Figure 4 can MISR be added? Again, specify which MODIS retrieval is used in the caption.
Figure 11e should there be some dashes, or are the cloud fraction changes nowhere statistically significant? If no changes are statistically significant it would be good to clarify that in the caption.
Figure 13 please specify what the contours are.

Can the two figures in S2 be put on the same color scale?

Caption of figure S3 – is it extinction, or change in extinction between SMK and CTL? Please spell this out

---

## Referee Comment (RC2) · Anonymous Referee #1 · 5 Aug 2020

This is a very interesting and important study topic. The manuscript described regional climate models simulation of biomass burning aerosol over southeast Atlantic, which draws very few attention in the literature but may have important influence due to the persistent intensive emission from South America. The modeling approach is reasonably, with solid validations and in-depth discussion of the result. The sensitivity simulations with different absorbing properties provided upper boundary estimates of the direct and semi-direct effect of aerosol. This is a well-organized study with fluent professional writing. Therefore I would recommend this manuscript to be accepted with

very minor revisions, following are some detailed comments.

Comment#1: The spatial distribution figures have very low DPI (although the information could be read), please make them more clear. Also, some figures have national boundaries but some don't, please keep it consistent. The curve figures have lines too slim, please consider make them bolder.

Comment#2: Line#23: Unnecessary to sate "the approach of using two ... of the results"

Comment#3: Line#35: the subsidence of air mass, water vapor, etc? please rephrase to be more clear

Comment#4: Line#36: so what is the overall semi-direct effect?

Comment#5: Line#39: "the results indicate ... to the absorbing properties of BBA" this is certainly true, please make more specific statement of the innovative finding from this study

Comment#6: Table 1. Horizontal resolution: 12km, 80km

Comment#7: Line#134: "In ALADIN-Climat ..." I don't understand this sentence, do you mean the boundary conditions were derived from simulations for a larger domain with biomass burning emission?

Comment#8: Line#189: Does CTL include direct and semi-direct effect of other aerosols?

Comment#9: Line#199: GFED gives fire emission as "dry matter" or "total carbon", what's the emission factors used to calculate aerosol emission?

Comment#10: Line#202 and section2.1.3: I am confused here, section2.1.3 mentioned BBA is treated as one type of aerosol in the model, so why the emission is upscaled for BC and OC separately?

[Figure]

Comment#11: Line#203: need reference for the scaling factor

Comment#12: Line#215: Raw GFED has 3-hour intervals.

Comment#13: Line#303: this section mainly described model evaluation of LCF, no detailed discussion was made regarding microphysical properties

Comment#14: section3.3.1: why AOD simulation bias is bigger in certain months, such as Jan-Apr and Sep-Dec; what's the correlation coefficient between simulation and satellite, with raw monthly data intervals?

Comment#15: Line#381-387: please provide more details to demonstrate the plume rise of biomass burning in the two models because it decides if BBA will get above or below cloud.

Comment#16: Fig5. The two model simulated different change of ACAOD from 2008 to 2009, please explain why

Comment#17: Line#446: prescribed SST can also be altered by the aerosol effect?

Comment#18: Fig.8: RegCM legend is vertical

Comment#19: Fig.11: why there are missing values?

---

## Author Comment (AC1) · 18 Sep 2020

*First of all, we would like to thank the reviewer for the various remarks. We have taken them into account in the new document.*

Anonymous Referee #1

This is a very interesting and important study topic. The manuscript described regional climate models simulation of biomass burning aerosol over southeast Atlantic, which draws very few attention in the literature but may have important influence due to the persistent intensive emission from South America. The modeling approach is reasonably, with solid validations and in-depth discussion of the result. The sensitivity simulations with different absorbing properties provided upper boundary estimates of the direct and semi-direct effect of aerosol. This is a well-organized study with fluent professional writing. Therefore I would recommend this manuscript to be accepted with very minor revisions, following are some detailed comments.

Comment#1: The spatial distribution figures have very low DPI (although the information could be read), please make them more clear. Also, some figures have national boundaries but some don't, please keep it consistent. The curve figures have lines too slim, please consider make them bolder.
*All the figures have been now improved following the different points indicated by the reviewer. National boundaries have been added for both figures and the size of the curves has been increased.*

Comment#2: Line#23: Unnecessary to sate "the approach of using two . . . of the results"
*This sentence is now removed.*

Comment#3: Line#35: the subsidence of air mass, water vapor, etc? please rephrase to be more clear
*This sentence has been improved in the new version.*

Comment#4: Line#36: so what is the overall semi-direct effect?
*This is a very interesting remark and we have now estimated the SDRE of BBA for the JAS period (2000-2015) for the two regional climate model. The following figure clearly indicates a negative (positive) SDRE where the low cloud fraction is increased (decreased) as shown in the Figure 11e and f. The important positive SDE over the Angola/Congo in ALADIN is due to changes in the high could fraction. For RegCM, a more uniform negative SDRE is obtained over SEA. In terms of magnitude, the SDRE is between ~-2 and -10 W m$^{-2}$ in RegCM over most of the SEA, higher that the mean value (-3.0 W.m$^{-2}$) reported by Sakaeda et al. (2011) at a climatic scale. All the points are now discussed in the new version (part 5.3) and the following figure has been added in the supplement material (Figure S7).*

[Figure]

**SDRE estimated by ALADIN (left) and RegCM (right) regional model (JAS season).**

Comment#5: Line#39: "the results indicate . . . to the absorbing properties of BBA" this is certainly true, please make more specific statement of the innovative finding from this study
*This is right and we have now included more statement on the DRE of BBA (at TOA) in the abstract using the following sentence : « Over the Sc region, DRE varies from +0.94 W m$^{-2}$ (scattering BBA) to +3.93 W m$^{-2}$ (most absorbing BBA)."*

Comment#6: Table 1. Horizontal resolution: 12km, 80km
*This is now changed in the Table 1.*

Comment#7: Line#134: "In ALADIN-Climat . . ." I don't understand this sentence, do you mean the boundary conditions were derived from simulations for a larger domain with biomass burning emission?
*This was effectively not clear, sorry. This sentence indicates that the ALADIN model is not forced at the lateral boundary by the long-range transport of aerosols. This means that, compared to RegCM, some bias in AOD could be due to the advection of particles that are not emitted directly in the ALADIN domain (see comment #14). We think the impact is minor as most of biomass-burning emission are included in the domain for the period studied here (JAS), but not necessary negligible.*

Comment#8: Line#189: Does CTL include direct and semi-direct effect of other aerosols?
*Yes, this important point is now indicated in the text.*

Comment#9: Line#199: GFED gives fire emission as "dry matter" or "total carbon", what's the emission factors used to calculate aerosol emission?
*In the RegCM and ALADIN simulations, we have directly used the emissions of BBA aerosol species already prepared in the CMIP6 dataset to force the two models at the surface. We have adjusted these emissions by using a scaling factor (1.5; similar for the two models) directly on the BC/OC emissions. The methodology used to derive and calculate the emissions is described in van Marle et al. 2017, which is referenced in the article.*

Comment#10: Line#202 and section2.1.3: I am confused here, section2.1.3 mentioned BBA is treated as one type of aerosol in the model, so why the emission is upscaled for BC and OC separately?

*In the models, the BC and OC GFED emission are used and merged to force the emission for the specific « smoke » tracer, which is then declined in fresh and aged BBA. This allows better comparisons with observations as mentioned in the article. In parallel and as used in the HadGEM model, a similar scaling factor is applied to BC and OC particles to reduce the bias with observed AOD (Thordnill et al. 2018).*

Comment#11: Line#203: need reference for the scaling factor
*The recent reference of Pan et al. (2020) has been added in the new version to highlight the fact that a large number of important scaling factors have been proposed for different emission datasets (GFED, QFED, FINN, GFAS and FEER).*

Comment#12: Line#215: Raw GFED has 3-hour intervals.
*This point has been precised. As the study is focused on climate simulations, we have effectively used monthly-mean emission data set and the diurnal cycle of smoke emission has not taken into account. This could impact the temporal variations of the aerosol loadings. This point is now mentioned in the text (2.1.4).*

Comment#13: Line#303: this section mainly described model evaluation of LCF, no detailed discussion was made regarding microphysical properties
*The cloud microphysical properties were not analysed in this study. In all simulations, we have fixed the cloud effective radius to 10 µm and the first indirect effect of BBA is absent in the two regional models. In that sense, we have focused our analyses on the LCF and LWP evaluation. However, it should be noted that the cloud optical depth (over the Sc region) has been validated in Mallet et al. (2019). These important points are now indicated in the text (2.1.4 and 3.2).*

Comment#14: section3.3.1: why AOD simulation bias is bigger in certain months, such as Jan-Apr and Sep-Dec; what's the correlation coefficient between simulation and satellite, with raw monthly data intervals?
*This is an interesting point and the differences detected in AOD during Jan-Apr and Sep-Dec periods could be due to different resasons as the long-range transport (especially for ALADIN that does not include chemical forcing at the boundaries) or some bias in the dynamic and the precipitation. In parallel, we can also note the high variability in the different products (reanalyses or remote-sensing) for these two seasons. For example, the two RCM are in a good agreement with MACv2 and MERRA data compared to MODIS and CMAS-RA. These points are now mentioned in the text and the temporal correlations with MODIS and MISR are now included in the Figure 4 for the two models. This shows a better agreement for RegCM (~0.95) compared to ALADIN (~0.80).*

Comment#15: Line#381-387: please provide more details to demonstrate the plume rise of biomass burning in the two models because it decides if BBA will get above or below cloud.
*This important point is indeed not discussed enough in the text and may explain some of the differences. The figure S3 indicates the BBA extinction (at 550 nm) and clearly shows an efficient transport of BBA between 1 and 4 km over the ocean in accordance with results of Das et al. (2017). This figure indicates also that the base of the smoke plume is lower in RegCM and may explain differences in ACAOD between the two regional models. This specific point is now clearly indicated in the new version in the part 3.3.2.*

**Das, S., Harshvardhan, H., Bian, H., Chin, M., Curci, G., Protonotariou, A. P., et al. (2017). Biomass burning aerosol transport and vertical distribution over the South African-Atlantic region. Journal of Geophysical Research.**

Comment#16: Fig5. The two model simulated different change of ACAOD from 2008 to 2009, please explain why

*The differences in ACAOD between the two RCMs are mainly due to the simulated AOD and the cloud top, which are respectively higher and lower in RegCM for these years, compared to ALADIN. This point is now added in the text.*

Comment#17: Line#446: prescribed SST can also be altered by the aerosol effect?
*This is effectively right as prescribed SST are also constructed using in-situ observations. This point is now mentioned in the new version (part 2.1.1).*

Comment#18: Fig.8: RegCM legend is vertical
*This is now changed in the new version.*

Comment#19: Fig.11: why there are missing values?
*The missing values are non-significant in the ALADIN model. This point is now clarified in the caption.*

---

## Author Comment (AC2) · 18 Sep 2020

Review of Mallet et al , Direct and semi-direct radiative forcing of biomass burning aerosols over the Southeast Atlantic (SEA) and its sensitivity to absorbing properties: a regional climate modeling study.

**First of all, we would like to thank the reviewer for the various remarks. We have taken them into account in the new document.**

Two atmospheric model simulations of direct and semi-direct effects of biomass burning aerosols over the south-east Atlantic ocean are compared. Simulations are performed over long enough periods to be climatically interesting. The main differences between models are that ALADIN uses prescribed SST and 12km horizontal resolution, RegCM does not and has 80km resolution. Both models use single-moment aerosol microphysics and neither represents indirect effects of aerosols on clouds. The model evaluation is excellent, with maximum use being made of the latest satellite products and detailed insights noted on the differences between various observational products. Biomass burning aerosol causes significant shortwave heating, consistently with other studies. The models significantly underestimate cloud fraction. This underestimate means they will be interesting to compare with other models that overestimate cloud fraction (eg Unified Model, Gordon et al 2018), but they get LWP about right – suggesting the clouds must also be too optically thick.

I think the topic is important, the analysis is sound and the paper is very well-written. It is well suited for publication in ACP and should be highly cited. My suggestions and comments, overall, are minor.

**Scientific comments**

Can you speculate further about why ALADIN-Climat underestimates the cloud fraction? Currently saying 'cloudiness, precipitation or boundary layer scheme' is a bit vague, though I appreciate the simulations are expensive and it may not be possible to do diagnostic sensitivity studies.

This underestimate of low cloud fraction in ALADIN over the southeast Atlantic is a feature shared with its global counterpart, namely ARPEGE-Climat, which is the atmospheric component of CNRM-CM6-1 (Roehrig et al. 2020). So far, both atmospheric model versions (regional and global) share the same code, the same parameterizations and the same tuning. In a recent study, Brient et al. (2019) analyse in depth the potential origin of the lack of clouds over the eastern parts of tropical oceans in ARPEGE-Climat. They found that these biases arise mostly from misrepresentation of subgrid effects on cloud formation (thus the cloud parameterization and its coupling with the turbulence parameterization) and partly from biases in drying induced by cloud-top entrainment mixing (turbulence parameterization). We believe that these results are applicable to ALADIN-Climat and therefore we now refer to this study in the manuscript.

**In that sense, we've included in the section 3.2 the following text :**

« This lack of LCF in ALADIN-Climat is consistent with the cloud biases found in its global counterpart (ARPEGE-Climat, Roehrig et al. 2020). Brient et al. (2019) attributed these biases to issues with the prescribed subgrid-scale distributions of water and temperature in the cloud parameterization and with and overestimated drying induced by the cloud-top entrainment parameterization. »

Roehrig, R., Beau, I., Saint Martin, D., Alias, A., Decharme, B., Guérémy, J.F., et al. (2020). The CNRM global atmosphere model ARPEGE Climat 6.3: Description and evaluation. Journal of Advances in Modeling Earth Systems, 12, e2020MS002075. https://doi.org/10.1029/2020MS002075.

Brient F., Roehrig, R., & Voldoire, A. (2019). Evaluating marine stratocumulus clouds in the CNRM-CM6-1 model using short-term hindcasts. Journal of Advances in Modeling Earth Systems, 11, 127–148, https://doi.org/10.1029/2018MS001461.

L340 can you dig a bit deeper into the huge difference between MODIS and MISR AOD? Could it be due to cloud masking? Maybe reference papers where the two retrievals have been compared to AERONET?

This is effectively an important issue that is still unresolved to our knowledge. We have already detected this significant difference between MODIS and MISR over this region in a previous study (Mallet et al., 2019). We have indicated that some of the land–ocean contrast in the satellite data comes from different factors, such as the over-land and over-water algorithms, which are different and may present different biases. The second is that cloud fraction is also significantly higher over the water than over the land, meaning that typically more days of data contribute to the monthly mean over land than over water.» We have now underlined all these important points in the new version (part 3.3.1).

In parallel, the AEROSAT community has also studied this issue and provides some interesting informations in the recent Sogacheva et al. (2020) overview. In this article, the MISR values are also systematically lower than the MODIS ones (better seen in the supplement material in Figure S7 for the SEA) but they don't highlight cloud-masking. In addition, one mentioned point is the following :"For the current MISR standard product, AOD is systematically underestimated for AOD > ~0.5. This is largely due to treatment of the surface boundary condition at high AOD (Kahn et al., 2010)". These points and the associated references are now included in the new version.

Sogacheva, L., Popp, T., Sayer, A. M., Dubovik, O., Garay, M. J., Heckel, A., Hsu, N. C., Jethva, H., Kahn, R. A., Kolmonen, P., Kosmale, M., de Leeuw, G., Levy, R. C., Litvinov, P., Lyapustin, A., North, P., Torres, O., and Arola, A.: Merging regional and global aerosol optical depth records from major available satellite products, Atmos. Chem. Phys., 20, 2031–2056, https://doi.org/10.5194/acp-20-2031-2020, 2020.

Kahn, R. A., Gaitley, B. J., Garay, M. J., Diner, D. J., Eck, T. F., Smirnov, A., and Holben, B. N.: Multiangle Imaging SpectroRadiometer global aerosol product assessment by comparison with the Aerosol Robotic Network, J. Geophys. Res., 115, D23209, https://doi.org/10.1029/2010JD014601, 2010.

Line 549- what about changes in inversion/cloud top height with smoke aerosol? Are there any such changes? Lines 568-580 say subsidence is reduced by smoke and tropospheric stability is decreased, but perhaps you can spell this out a bit more? Dipole patterns in Figures 12 (bottom-right) and S4 suggest cloud height changes might be occurring.

This is a very interesting question and we have now included the analyse of the changes between the SMK and CTL runs for the cloud top height (in Pascal, see the following figure). We can observe that the cloud top heigth is increased (by ~30hPa) in the SMK simulation. This point is now included in the text (part 5.2). Due to the number of figures, we include this new figure in the supplement document (Figure S5).

One other thing missing from the analysis is an evaluation of free-tropospheric (and boundary layer) relative humidity. Do the models replicate the observed increases in RH associated with, or coincident with, smoke layers?

This is an important point that we have detailled in a previous study (Mallet et al., 2019) focused on the evaluation of the ALADIN-Climate model during the ORACLES-2016 campaign. In this study, we found that the model suffers to simulate the relative humidity within the smoke plume (see the following figure from Mallet et al., ACP 2019). In its nudged version (ALADIN\_RH(SN)), the RH biais is considerably reduced in ALADIN. This could affect the optical properties of smoke and the AOD. We have now mentioned this important point in the article (Part 3.3.1) to underline that some biais in AOD could be related to the RH underestimate.

Relative humidity profiles from non-nudged and nudged (SN) ALADIN simulations compared to MERRA2 data. (Figure 12 from Mallet et al., ACP 2019)

In addition and at a climatic scale, the following figure indicates that the ALADIN-Climate model is able to represent the main relative humidity regional pattern both in the BL and FT for the JAS season (2000-2015 period).

Relative humidity simulated by ALADIN-Climat (left) and obtained from ERA-INT (right) for the JAS season and 2000-2015 period (up at 900 hPa and 700hPa below)

Finally, I was expecting to see a discussion of the net semi-direct radiative effect of the BBA in Wm -2 . If the DRE is (F-F clean ) SMK -(F-F clean ) CTL , as per Ghan (2013), can the SDRE be calculated in these simulations as F clean, CTL -F clean, SMK ? (not sure I got the sign right, but you get the idea). We would presumably expect RegCM to show a negative SDRE as Figure 11 shows an increase in cloud fraction (albeit perhaps not statistically significant) and Figure 12 increases in LWC, while ALADIN will have a positive SDRE in some regions.

This is a very interesting remark and we have now estimated the SDRE of BBA for the JAS period (2000-2015) for the two regional climate model. The following figure clearly indicates a negative (positive) SDRE where the low cloud fraction is increased (decreased) as shown in the Figure 11e and f. The important positive SDE over the Angola/Congo in ALADIN is due to changes in the high cloud fraction. For RegCM, a more uniform negative SDRE is obtained over SEA as noted by the reviewer. In terms of magnitude, the SDRE is between ~-2 and -10 W m-2 in RegCM over most of the SEA, higher that the mean value ( $-3.0 W m^{-2}$ ) reported by Sakaeda et al. (2011) at a climatic scale. All those points are now included in the new version (part 5.3) and the following figure has been added in the supplement material (Figure S7).

---

## Author Comment (AC3) · 18 Sep 2020

Review of Mallet et al , Direct and semi-direct radiative forcing of biomass burning aerosols over the Southeast Atlantic (SEA) and its sensitivity to absorbing properties: a regional climate modeling study.

*First of all, we would like to thank the reviewer for the various remarks. We have taken them into account in the new document.*

Two atmospheric model simulations of direct and semi-direct effects of biomass burning aerosols over the south-east Atlantic ocean are compared. Simulations are performed over long enough periods to be climatically interesting. The main differences between models are that ALADIN uses prescribed SST and 12km horizontal resolution, RegCM does not and has 80km resolution. Both models use single-moment aerosol microphysics and neither represents indirect effects of aerosols on clouds. The model evaluation is excellent, with maximum use being made of the latest satellite products and detailed insights noted on the differences between various observational products. Biomass burning aerosol causes significant shortwave heating, consistently with other studies. The models significantly underestimate cloud fraction. This underestimate means they will be interesting to compare with other models that overestimate cloud fraction (eg Unified Model, Gordon et al 2018), but they get LWP about right – suggesting the clouds must also be too optically thick.

I think the topic is important, the analysis is sound and the paper is very well-written. It is well suited for publication in ACP and should be highly cited. My suggestions and comments, overall, are minor.

Scientific comments

Can you speculate further about why ALADIN-Climat underestimates the cloud fraction? Currently saying 'cloudiness, precipitation or boundary layer scheme' is a bit vague, though I appreciate the simulations are expensive and it may not be possible to do diagnostic sensitivity studies.

*This underestimate of low cloud fraction in ALADIN over the southeast Atlantic is a feature shared with its global counterpart, namely ARPEGE-Climat, which is the atmospheric component of CNRM-CM6-1 (Roehrig et al. 2020). So far, both atmospheric model versions (regional and global) share the same code, the same parameterizations and the same tuning. In a recent study, Brient et al. (2019) analyse in depth the potential origin of the lack of clouds over the eastern parts of tropical oceans in ARPEGE-Climat. They found that these biases arise mostly from misrepresentation of subgrid effects on cloud formation (thus the cloud parameterization and its coupling with the turbulence parameterization) and partly from biases in drying induced by cloud-top entrainment mixing (turbulence parameterization). We believe that these results are applicable to ALADIN-Climat and therefore we now refer to this study in the manuscript.*

*In that sense, we've included in the section 3.2 the following text :*
*« This lack of LCF in ALADIN-Climat is consistent with the cloud biases found in its global counterpart (ARPEGE-Climat, Roehrig et al. 2020). Brient et al. (2019) attributed these biases to issues with the prescribed subgrid-scale distributions of water and temperature in the cloud parameterization and with and overestimated drying induced by the cloud-top entrainment parameterization. »*

*Roehrig, R., Beau, I., Saint Martin, D., Alias, A., Decharme, B., Guérémy, J.F., et al. (2020). The CNRM global atmosphere model ARPEGE Climat 6.3: Description and evaluation. Journal of Advances in Modeling Earth Systems, 12, e2020MS002075. https://doi .org/ 10.1029/2020MS002075.*

*Brient F., Roehrig, R., & Voldoire, A. (2019). Evaluating marine stratocumulus clouds in the CNRM-CM6-1 model using short-term hindcasts. Journal of Advances in Modeling Earth Systems, 11, 127–148, https://doi.org/10.1029/2018MS001461.*

L340 can you dig a bit deeper into the huge difference between MODIS and MISR AOD? Could it be due to cloud masking? Maybe reference papers where the two retrievals have been compared to AERONET?

*This is effectively an important issue that is still unresolved to our knowledge. We have already detected this significant difference between MODIS and MISR over this region in a previous study (Mallet et al., 2019). We have indicated that some of the land–ocean contrast in the satellite data comes from different factors, such as the over-land and over-water algorithms, which are different and may present different biases. The second is that cloud fraction is also significantly higher over the water than over the land, meaning that typically more days of data contribute to the monthly mean over land than over water.» We have now underlined all these important points in the new version (part 3.3.1).*

*In parallel, the AEROSAT community has also studied this issue and provides some interesting informations in the recent Sogacheva et al. (2020) overview. In this article, the MISR values are also systematically lower than the MODIS ones (better seen in the supplement material in Figure S7 for the SEA) but they don't highlight cloud-masking. In addition, one mentioned point is the following :"For the current MISR standard product, AOD is systematically underestimated for AOD>~0.5. This is largely due to treatment of the surface boundary condition at high AOD (Kahn et al., 2010)". These points and the associated references are now included in the new version.*

*Sogacheva, L., Popp, T., Sayer, A. M., Dubovik, O., Garay, M. J., Heckel, A., Hsu, N. C., Jethva, H., Kahn, R. A., Kolmonen, P., Kosmale, M., de Leeuw, G., Levy, R. C., Litvinov, P., Lyapustin, A., North, P., Torres, O., and Arola, A.: Merging regional and global aerosol optical depth records from major available satellite products, Atmos. Chem. Phys., 20, 2031–2056, https://doi.org/10.5194/acp-20-2031-2020, 2020.*

*Kahn, R. A., Gaitley, B. J., Garay, M. J., Diner, D. J., Eck, T. F., Smirnov, A., and Holben, B. N.: Multiangle Imaging SpectroRadiometer global aerosol product assessment by comparison with the Aerosol Robotic Network, J. Geophys. Res., 115, D23209, https://doi.org/10.1029/2010JD014601, 2010.*

Line 549- what about changes in inversion/cloud top height with smoke aerosol? Are there any such changes? Lines 568-580 say subsidence is reduced by smoke and tropospheric stability is decreased, but perhaps you can spell this out a bit more? Dipole patterns in Figures 12 (bottom-right) and S4 suggest cloud height changes might be occurring.

*This is a very interesting question and we have now included the analyse of the changes between the SMK and CTL runs for the cloud top height (in Pascal, see the following figure). We can observe that the cloud top heigth is increased (by ~30hPa) in the SMK simulation. This point is now included in the text (part 5.2). Due to the number of figures, we include this new figure in the supplement document (Figure S5).*

One other thing missing from the analysis is an evaluation of free-tropospheric (and boundary layer) relative humidity. Do the models replicate the observed increases in RH associated with, or coincident with, smoke layers?

*This is an important point that we have detailled in a previous study (Mallet et al., 2019) focused on the evaluation of the ALADIN-Climate model during the ORACLES-2016 campaign. In this study, we found that the model suffers to simulate the relative humidity within the smoke plume*

*(see the following figure from Mallet et al., ACP 2019). In its nudged version (ALADIN_RH(SN)), the RH biais is considerably reduced in ALADIN. This could affect the optical properties of smoke and the AOD. We have now mentioned this important point in the article (Part 3.3.1) to underline that some biais in AOD could be related to the RH underestimate.*

[Figure]

*Relative humidity profiles from non-nudged and nudged (SN) ALADIN simulations compared to MERRA2 data. (Figure 12 from Mallet et al., ACP 2019)*

*In addition and at a climatic scale, the following figure indicates that the ALADIN-Climate model is able to represent the main relative humidity regional pattern both in the BL and FT for the JAS season (2000-2015 period).*

[Figure]

*Relative humidity simulated by ALADIN-Climat (left) and obtained from ERA-INT (right) for the JAS season and 2000-2015 period (up at 900 hPa and 700hPa below)*

Finally, I was expecting to see a discussion of the net semi-direct radiative effect of the BBA in Wm -2 . If the DRE is (F-F clean ) SMK -(F-F clean ) CTL , as per Ghan (2013), can the SDRE be calculated in these simulations as F clean,CTL -F clean,SMK ? (not sure I got the sign right, but you get the idea). We would presumably expect RegCM to show a negative SDRE as Figure 11 shows an increase in cloud fraction (albeit perhaps not statistically significant) and Figure 12 increases in LWC, while ALADIN will have a positive SDRE in some regions.

*This is a very interesting remark and we have now estimated the SDRE of BBA for the JAS period (2000-2015) for the two regional climate model. The following figure clearly indicates a negative (positive) SDRE where the low cloud fraction is increased (decreased) as shown in the Figure 11e and f. The important positive SDE over the Angola/Congo in ALADIN is due to changes in the high cloud fraction. For RegCM, a more uniform negative SDRE is obtained over SEA as noted by the reviewer. In terms of magnitude, the SDRE is between ~-2 and -10 W m$^{-2}$ in RegCM over most of the SEA, higher that the mean value ($-3.0$ W m$^{-2}$) reported by Sakaeda et al. (2011) at a climatic scale. All those points are now included in the new version (part 5.3) and the following figure has been added in the supplement material (Figure S7).*

[Figure]

*SDRE estimated by ALADIN (left) and RegCM (right) models (JAS season and 2000-2015 period).*

Text comments and suggestions

Line 300: missing "such" in "such as"?
*Yes, this is now modified in the text.*

The text says both models are "driven" by ERA-interim reanalysis – does that mean the boundary conditions are derived from ERA-interim or is there nudging of horizontal winds?How do the two models handle sub-grid cloud?

*For the two models, there is no nudging and they are effectively « driven » by ERA-Inetrim as the boundary conditions. Concerning the sub-grid clouds, ALADIN uses a cloud parametrization based on the description of the statistical joint distribution of total water and liquid potential temperature, following Somerria and Deardorff (1977) and Bougeault (1981). The subgrid-scale variances of the latter variables are diagnosed using information from the turbulence scheme (Cuxart et al., 2000). The scheme is fully describd in Roehrig et al. (2020). In RegCM, the convective cloud fraction is parametrized according to selected convective schemes, while cloud water content is estimated depending on a temperature based parametrisation (Giorgi et al., 2012). Subgrid cloud fractions and cloud water content are combined to resolved cloud fraction and water content before being passed to the radiation scheme. All these points are now detailed in the new version (see section 2.1.1).*

*Bougeault, P. (1981). Modeling the Trade-Wind Cumulus Boundary Layer. Part I: Testing the Ensemble Cloud Relations Against Numerical Data. Journal of the Atmospheric Sciences, 38(11), 2414–2428.*

*Cuxart, J., Bougeault, P., & Redelsperger, J. L. (2000). A turbulence scheme allowing for mesoscale and large-eddy simulations. Quarterly Journal of the Royal Meteorological Society, 126(562), 1–30. https://doi.org/10.1002/qj.49712656202*

*Giorgi, F., Coppola, E., Solmon, F., Mariotti, L., Sylla, M., Bi, X., et al.: RegCM4: model description and preliminary tests over multiple CORDEX domains, Clim. Res., 52, 7–29, 2012.*

*Sommeria, G., & Deardorff, J. W. (1977). Subgrid-Scale Condensation in Models of Nonprecipitating Clouds. Journal of the Atmospheric Sciences, 34(2), 344– 355.*

What does MACv2 do for the cloud fraction/water path? Or what is used for the cloud fraction in the calculation to get the DRE from MACv2 on the right of Figure 8?
**The radiative transfer method described in Kinne (2019) uses the ISCCP-based cloud cover for high (<440 hPa), middle (440-680 hPa) and low (> 680 hPa) altitudes. This clarification is now indicated in the caption of the Figure 8.**

Models assume external mixing of aerosols. Both represent fresh and aged smoke, but there is not a separation between BC and OC. This is interesting and complementary to other models. Both use scaled GFED emissions (could additionally cite https://www.atmos-chem-phys.net/20/969/2020/ to justify this). Which version of GFED?
***The Pan et al. (2020) paper has been added in the new version. For all the simulations, the GFED version 4 (Marle et al., 2017) has been used, which are also those used in the CMIP6 modeling exercice. These points are now indicated in the text.***

Line 460 When reading this, at first I got slightly confused between decreases in cloud fraction in SMK compared to CTL (which are irrelevant here) and decreases in cloud fraction from south to north, which is what you are talking about. Maybe rephrase to "decrease in low cloud fraction with latitude as one moves northwards from 5S" or similar.
***This is now changed in the new version.***

Line 495 CTRL->CTL.
***This is now changed in the text.***

Line 668 would be good to put some numbers on the TOA DRE as you do for the surface DRE.
***The TOA DRE of smoke is now documented over the main Stratocumulus region in the conclusion.***

Figure 1 would be helpful to reference Klein and Hartmann in the caption, as in Figure 2.
***This is now indicated in the Figure 1.***

Figure 2 can yellow be replaced by orange for RegCM_CTL, so it doesn't fade into the background?
***This is now changed in the Figure 2.***

Figure 2b do you show grid-box-average or in-cloud LWP?
***This is effectively an important point and we have indicated the grid-box mean in the Figure 2b. This is is now indicated in the figure caption,***

Figure 3 specify that (if I am correct) this is MODIS standard AOD, not MODIS ACAOD.
***This is effectively MODIS standard AOD data. This is now indicated in the Figure 3 caption.***

Figure 4 can MISR be added? Again, specify which MODIS retrieval is used in the caption.
***MISR is now added in the Figure 4 for the 2001-2015 period. The MODIS AOD data are specified in the caption.***

Figure 11e should there be some dashes, or are the cloud fraction changes nowhere statistically significant? If no changes are statistically significant it would be good to clarify that in the caption.
***This was a mistake, sorry. The new figure 11e is now updated in the new version, including the t-test.***

Figure 13 please specify what the contours are.
***This is now done.***

Can the two figures in S2 be put on the same color scale?
***This is done.***

Caption of figure S3 – is it extinction, or change in extinction between SMK and CTL? Please spell this out
***This is effectively the BBA extinction profiles at 550 nm. This is now modified.***